

# Psycho-physio-neurological correlates of qualitative attention, emotion and flow experiences in a close-to-real-life extreme sports situation: low- and high-altitude slackline walking

Marcelo Felipe de Sampaio Barros[1,2], Carlos Alberto Stefano Filho[3,4], Lucas Toffoli de Menezes[3,4], Fernando Manuel Araújo-Moreira[1,5], Luis Carlos Trevelin[1,6], Rafael Pimentel Maia[7], Rémi Radel[2] and Gabriela Castellano[3,4]

[1] Programa de Pós-graduação em Biotecnologia, Universidade Federal de São Carlos (UFSCar), São Carlos, São Paulo, Brazil
[2] Laboratoire LAMHESS, Université de Nice Sophia Antipolis, Nice, Côte d'Azur, France
[3] Neurophysics Group, Gleb Wataghin Institute of Physics, Universidade Estadual de Campinas, Campinas, São Paulo, Brazil
[4] Brazilian Institute of Neuroscience and Neurotechnology (BRAINN), Campinas, São Paulo, Brazil
[5] Programa de pós-graduação em Engenharia Nuclear, Instituto Militar de Engenharia/IME, Rio de Janeiro, Rio de Janeiro, Brazil
[6] Departamento de Computação, Universidade Federal de São Carlos (UFSCar), São Carlos, São Paulo, Brazil
[7] Department of Statistics, Institute of Mathematics, Statistics and Scientific Computing, Universidade Estadual de Campinas, Campinas, São Paulo, Brazil

Corresponding authors
Marcelo Felipe de Sampaio Barros, marcelofelipebarros@hotmail.com
Gabriela Castellano, gabriela@ifi.unicamp.br

## ABSTRACT

It has been indicated that extreme sport activities result in a highly rewarding experience, despite also providing fear, stress and anxiety. Studies have related this experience to the concept of flow, a positive feeling that individuals undergo when they are completely immersed in an activity. However, little is known about the exact nature of these experiences, and, there are still no empirical results to characterize the brain dynamics during extreme sport practice. This work aimed at investigating changes in psychological responses while recording physiological (heart rate–HR, and breathing rate–BR) and neural (electroencephalographic–EEG) data of eight volunteers, during outdoors slackline walking in a mountainous environment at two different altitude conditions (1 m–low-walk– and 45 m–high-walk–from the ground). Low-walk showed a higher score on flow scale, while high-walk displayed a higher score in the negative affect aspects, which together point to some level of flow restriction during high-walk. The order of task performance was shown to be relevant for the physiological and neural variables. The brain behavior during flow, mainly considering attention networks, displayed the stimulus-driven ventral attention network–VAN, regionally prevailing (mainly at the frontal lobe), over the goal-directed dorsal attention network–DAN. Therefore, we suggest an interpretation of flow experiences as an opened attention to more changing details in the surroundings, *i.e.*, configured as a 'task-constantly-opened-to-subtle-information experience', rather than a 'task-focused experience'.

## INTRODUCTION

In the last decades, extreme sports which imply risk, such as rock-climbing in hazardous environments, mountaineering, base jumping, or other physical activities with odds of deadly falls, have become extremely popular (*Brymer et al., 2020*; *Kerr & Houge Mackenzie, 2012*; *Lee et al., 2024*; *Pizam, Reichel & Uriely, 2001*). While this practice has been traditionally associated with people with a high sensation-seeking personality (*e.g.*, *Pizam, Reichel & Uriely, 2001*), recent conceptualization indicates that participation in these activities would extend beyond this restrictive profile by satisfying many different motives (*Clough et al., 2016*; *Kerr & Houge Mackenzie, 2012*) and by providing a highly rewarding experience (*Immonen et al., 2017*). Because high-risk sports typically also imply fear, stress, and anxiety (*Hare, Wetherell & Smith, 2013*; *Monasterio et al., 2016*; *Woodman et al., 2009*), the positive experience must be very intense to prevail over these negative emotions. However, little is known about the exact nature of these experiences.

Studies investigating psychophysiological changes associated with extreme sport have indicated that some alterations are triggered before practice. Marked increases in self-reported anxiety and cortisol are typically seen at the resting period preceding the practice (*Chatterton et al., 1997*; *Hare, Wetherell & Smith, 2013*; *Mujica-Parodi et al., 2014*), peaking during practice (*Hare, Wetherell & Smith, 2013*; *Mujica-Parodi et al., 2014*). Interestingly, the amplitude of the cortisol response in skydiving was found to be highly correlated with amygdala reactivity in a separate threat perception task (*Mujica-Parodi et al., 2014*). Besides the hypothalamic–pituitary–adrenal axis activation, the autonomous nervous system is also importantly affected during extreme sports practice. Physiological signals (usually accessed by wearable devices), can reflect people's emotions. Indeed, emotions influence the activity of the autonomous nervous system (ANS), which in turn regulates various body parameters such as heart rate, respiration and temperature pattern variations. Besides of the wearable devices, contactless technologies, such thermal infrared (IR) imaging based on machine learning approaches are starting to be used for the assessment of emotions (*Filippini et al., 2022*). Skydiving jump leads to increased sympathetic activity as indexed by higher salivary alpha-amylase (*Chatterton et al., 1997*) or heart rate (HR) (*Allison et al., 2012*), which suggests a state of hyperarousal (*Sterlini & Bryant, 2002*). However, if the activation of the sympathetic system can be part of the stress response, results showing dissociation of the sympathetic arousal from cortisol reactivity (*Monasterio et al., 2016*) suggest the involvement of other factors, such as emotional excitement (*Shiota et al., 2011*), or the activation of the brain reward system (*Critchley, Mathias & Dolan, 2001*). Supporting this idea, it has been shown that risk sensation seeking is related to dopaminergic activity (*German & Bowden, 1974*), and that extreme sports could lead to similar effects to those of drugs that stimulate the reward system, contributing to anhedonia (*Franken, Zijlstra & Muris, 2006*) and to a state of dependency (*Partington, Partington & Olivier, 2009*). It is interesting to note that the sympathetic

activation during extreme sport practice does not necessarily coincide with parasympathetic deactivation, but, in contrast, the activations of the two autonomous nervous branches appear to occur in a concurrent way at the same time. Specifically, skydiving has been found to elevate both HR, a sympathetic signal, and heart rate variability (HRV), a parasympathetic acting indication (*Allison et al., 2012*). Parasympathetic activation should help individuals to control typical anxiety effects during extreme sports. Co-activation of the two autonomous nervous branches would thus lead to a control sensation state with high awake and elevated calmness (*Engelbregt et al., 2022*) often bringing experiences of 'relaxed subjective tension' in action during risk situations. Such simultaneous sympathetic/parasympathetic co-activation has been related to the concept of flow experiences (*Harmat et al., 2015*; *Ullén et al., 2010*), a positive feeling that individuals experience when they are completely immersed in an activity (*Polechoński et al., 2024*; *Csikszentmihalyi, 2014*). Recent conceptualization of flow describes it as a state of effortless attention (*Bruya, 2010*), which could suggest flow as 'relaxed high attention' experiences. Many accounts have proposed that extreme sports would represent a perfect situation to induce flow (*Jackson & Csikszentmihalyi, 1999*; *Pain & Pain, 2005*). Flow states occur suddenly at critical time points (*Weber & Fisher, 2020*). Surgery and mountain climbing are highly critical tasks, which are more often reported to result in intense, ecstatic flow experiences, whereas yet absorbing but less critical tasks such as reading and video games, have less intense flow feelings (*Gold & Ciorciari, 2020*). Nevertheless, despite *Gold & Ciorciari*'s *(2020)* opinion in their article, however, a person can experience intense flow also during a video game playing (*e.g.*, an expert gamer playing a new game). This will depends on the person and on their activity. The fact is that extreme sports provide alternative perceptions of space, time, and clarity or an augmented state of sensual awareness, that seem to facilitate flow, and, when in turn that happens, it tends to enhance performers' ability to act (*Brymer & Schweitzer, 2017*). The study of the psychophysiological changes associated with the practice of extreme sports, reflecting on autonomic nervous systems behavior, suggests that this practice leads to a unique and complex experience that combines aspects of stress, anxiety, arousal, excitement, reward, and concentration (emotion and attention qualities), that can culminate in flow experiences. Since these elements recruit different neural networks, which are sometimes seem to work more in opposition than in congruence (*Hermans et al., 2014*), it would be interesting to better examine the neural patterns occurring during extreme sport performance. Studies about emotion (*Doufesh et al., 2014*), attention (*Jäncke, Leipold & Burkhard, 2018*), or both (*Engelbregt et al., 2022*), have shown autonomic nervous system body responses with relation to electroencephalography (EEG) measures.

Several studies have suggested that emotional perception tends to be brain lateralized by valence (*Mańkowska et al., 2018*; *Martin & Altarriba, 2017*). The hemispheric laterality valence hypothesis (*Hellige, 1993*) proposes that the right hemisphere is dominant for processing negative, unpleasant emotions, with the left hemisphere being dominant for positive, pleasant ones. Attention is one of the most studied cognitive processes in the social sciences. It is also one of the most difficult processes to define. There is no single

definition of attention, but it is largely conceptualized as the means by which the brain chooses information (sensorial or from previously formed mental representations) for further processing (*Weber et al., 2009*). Attention considering brain networks' connectivity and its variations, has been studied using cross-talk simultaneous EEG-fMRI technics (*Walz et al., 2013*). Psycho-neuro-physiological mechanisms associated with flow experiences have started to be elucidated (*de Sampaio Barros et al., 2018*; *Yu et al., 2023*), but its underlying neural correlates are yet to be better studied and established. Attempts to explain flow experiences in terms of brain connectivity networks have used theories such as the Multiple Demand (MD) network (*Harris, Vine & Wilson, 2017b*). Another theories, such as the transient hypofrontality hypothesis (*Dietrich, 2003*) and the synchronization theory of flow (STF) (*Huskey, Wilcox & Weber, 2018*; *Weber et al., 2009*; *Weber & Fisher, 2020*) were specifically developed to suggest how the brain could behave during flow. The MD theory, as well as the STF, suggests that flow results from larger activation of the focused attentional control neural network, involving mainly the dorsal attention network (DAN), with an inhibition of the salience-driven attention network, which mainly encompasses the ventral attention network (VAN), in order to prevent reorienting of attention to salient task-irrelevant stimuli (*Gold & Ciorciari, 2021*). As dynamic structures, brain networks present coalitions and overlapping interconnections (*Pessoa, 2017*, *2018*), with inputs and outputs of any given region coming from, and relaying to, several cortical and subcortical areas. Furthermore, a given brain region can be affiliated to more than one network, shifting from one to another depending on task demands (*Pessoa, 2018*). Thus, to get a fuller understanding of neural circuit functions, it is necessary to consider both local and large-scale circuit interactions (*Pessoa, 2017*).

The aim of the present study was to explore changes in the psychological (qualitative attention, emotion and subjective flow experiences), peripheral-physiological (heart rate and breathing rate), and neuro-physiological (EEG analysis of power spectral density and functional connectivity), responses during a close-to-real-life of a high-risk sport situation, offering an empirical description of these intriguing experiences. Although brain functional connectivity has been more commonly investigated using fMRI approaches, we chose a portable EEG system to record neurophysiological signals, since this experiment, in a real extreme sport environment, would have been impossible to perform with fMRI. A study was conducted using EEG mobile equipment collecting brain responses during one tightrope walk at 15 m altitude (*Leroy & Cheron, 2020*). They used a swLORETA brain source location protocol to study brain dynamics differences between flow and stressful situations during walking. We did not perform source location since this approach relies on estimates of head shape, brain composition and different conductivity across tissues, which are difficult to obtain with high accuracy unless the corresponding MRI is available (*Eom, 2023*), which was not the case here. As scalp EEG analysis is simpler and its interpretation is more straight-forward, we opted to do the data analysis based on previous literature that could allow to consider the EEG electrode sites with more propensity to pick up signals on the scalp arising from specific brain networks described by pioneering fMRI technics (*Luo et al., 2022*; *Rojas et al., 2018*; *Walz et al., 2013*).

We chose a slackline walking protocol with varied altitudes, a high- (45 m) and a low-walk (1 m). In real life, some slackliners do the high altitude walk without safety ropes, thus configuring an imminent life-threatening situation. Our study protocol, however, did not configure a real risk situation, as participants were attached to safety ropes. Nevertheless, we tried to play-act a close to real threatening situation, exploring subjective feelings of risk according to the height of the line as a way to elicit flow experiences. The slackline, this modern form of funambulism, which has gained increased popularity over the last decade, was chosen for two reasons: first, because this activity allowed a well-controlled experimental design by manipulating the subjective feeling of risk according to the height of the line, while keeping all other parameters of performance constant; second, because participants typically move slowly, limiting the amount of motion induced-artifacts in the EEG recordings, same reason why the protocol was chosen for the tightrope study (*Leroy & Cheron, 2020*).

Some studies can be found in the literature addressing slackline practice, investigating the balancing capabilities of experts and beginners (*Stein & Mombaur, 2022*), or the metabolic demands of less and more experienced practitioners (*Ueta et al., 2022*). A study using a slackline walking task and subjective informed awareness, reconceptualizes flow experiences as a property of the self-perception performer-environment coupling (*Montull et al., 2020*). Other work, looked at functional connectivity related to slacklining (*Baláš et al., 2023*), but they used fMRI, which means that measurements were performed before and after the actual practice, and not during, as in the present work. Another work which has recorded brain changes during the slacklining practice used a fNIRS equipment (*Seidel-Marzi et al., 2021*), but they did not study the brain in highline situations.

## MATERIALS AND METHODS

### Subjects

Eight adult volunteers (all males; age: 27 ± 7 years) without history of brain injury were included in the experiment. Inclusion criteria were: 1. to hold extensive slackline practice; and 2. to have already tried highline walking. All participants gave their written informed consent and were free to withdraw from the experiment at any time. The study's protocol was approved by the scientific council of the University (CER–Comité dÉthique de la Recherche d'Université Côte d'Azur under n° 2023-002), and was performed in accordance with the ethical standards of the declaration of Helsinki.

### General procedure

The experiment was conducted outdoors in a mountainous environment next to the city of Nice, in the French Alps. This place is composed of steep rocky limestone cliffs falling straight into the Mediterranean Sea and the city of Monaco.

Participants were first equipped with a security harness, a thoracic belt collecting cardiac and respiratory measures, the EEG headcap, and a small backpack containing the EEG data acquisition system. Then, participants were asked to complete two slackline walks on two separate lines. For each line, the given oral instructions were to traverse the line entirely and return. By essence, the implicit goal was to complete this objective while also

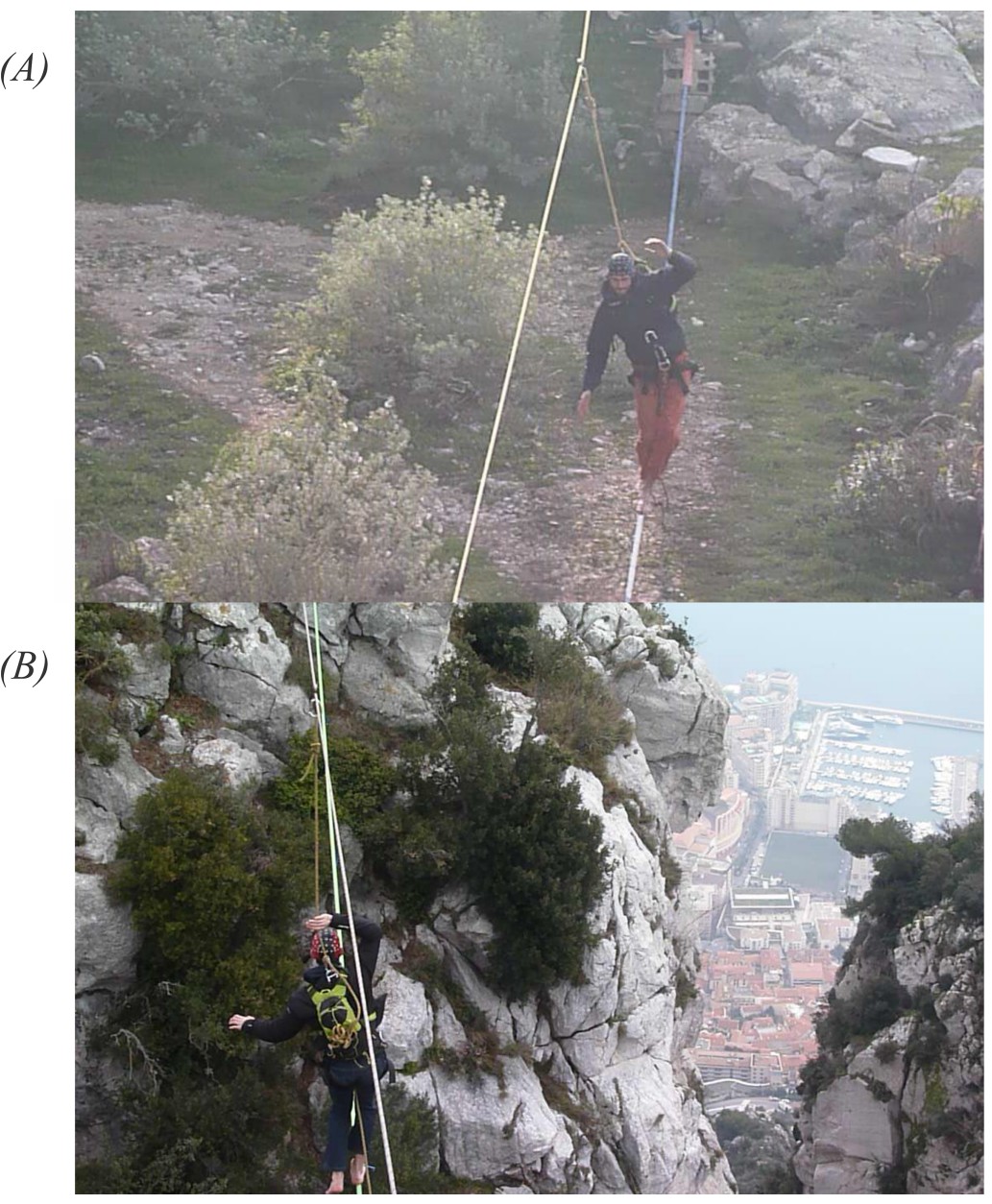

(A)

(B)

**Figure 1 Participants walking in the slackline tapes.**

minimizing the number of falls. The two lines were similar regarding the important parameters that influence the objective difficulty of the task. Specifically, both lines had the same 22 m length, were made from the same 5 cm width polyester sling, and were attached with the same equipment at the same tension. Above the two slacklines, a static security rope was firmly attached to prevent participants from making long falls using a safety leash connected to the back of the safety harness (Fig. 1). This safety system is somewhat different from usual practices: for the slackline at the ground, subjects normally use no security systems. In highlines, the safety rope is usually strapped just under the walk line, which leads to longer and more impressive falls as the participants fall upside down. Our

different security system was chosen to limit possible damage to the equipment and to standardize conditions between the two situations. The two lines differed only in terms of the height at which they were set. One line was attached just above the ground at a height of about 1.2 m (low-walk). Note that, due to elasticity, the distance from the ground could be reduced to only 20 cm at the midpoint, when the line was loaded with the participant's weight. The ground was flat and cleared from any obstacle. For the highline, the line was 45 m above a steep gully descending straight to the seaside, giving the visual impression of a 500 m height (high-walk) (Fig. 1B). The two lines were distant by about 500 m from each other. The order of the walking condition was counterbalanced across participants. Before each walking condition, participants sat quietly for 2 min and the last 30 s counted as the preliminary rest period. Immediately after exiting each line, subjects were asked to complete a psychological questionnaire.

The whole experiment was video-recorded to mark periods of interest in the recorded biosignals (heart rate, breathing rate, EEG). During the two walking conditions, only the signals recorded when participants were able to properly stand on the line (*i.e.*, without holding the leash) were retained for analysis.

## Psychological measures

### Flow experience

The short form of the Flow State Scale 2–FSS-2 (*Jackson, Martin & Eklund, 2008*), rated on a 7-point Likert scale was included in the questionnaire, filled after each walking condition (Cronbach $\alpha$ = 0.780). The authors have permission to use this instrument from the copyright holders. This scale assesses the nine components, or key characteristics, proposed in the conceptual theory of flow (*Csikszentmihalyi, 1990*), which are evaluated by the answers of: "1-I felt I was competent enough to meet the high demands of the situation" (Challenge-Skill Balance); "2-I did things spontaneously and automatically without having to think" (Action-Awareness Merging); "3-I had a strong sense of what I want to do" (Clear Goals); "4-I had a good idea while I was performing about how well I was doing" (Unambiguous Feedback); "5-I was completely focused on the task at hand" (Concentration); "6-I had a feeling of total control over what I was doing" (Sense of Control); "7-The way time passed seemed to be different from normal" (Transformation of Time); "8-The experience was extremely rewarding" (Autotelic Experience); "9-I was not worried about what others may have been thinking of me or my performance" (Loss of Self-Consciousness). The answers to item 7 (Transformation of Time) led to a large decrement of the internal scale consistency, decreasing Cronbach $\alpha$ from 0.780 to 0.517; hence, as done in previous studies (*e.g.*, *Scott-Hamilton, Schutte & Brown, 2016*), this item was excluded for further analysis when calculating the composite scores. The answers of the French translations to the other eight items were considered for statistical analysis in composite scores. Since recent conceptualization of flow describes it as a state of effortless attention (*Bruya, 2010*), two questions were added to measure the extent to which participants were able to maintain their focus of attention, and to measure the amount of subjective effort required to focus their attention ("10. I was very concentrated during the task" and "11. There was no effort to concentrate"). These items have been used in several

studies (*e.g.*, *Csikszentmihalyi & Nakamura, 2010*; *Harmat et al., 2015*). Items 10 (named attention) and 11 (named effortless attention), were analyzed separately from the FSS-2.

### Positive and negative affects

A part of the psychological questionnaire assessed participants' positive and negative aspects using the French version of the Short-Form of the Positive and Negative Affect Schedule (PANAS), (*Gaudreau, Sanchez & Blondin, 2006*; *Thompson, 2007*). This form includes two 5-item scales to measure both positive and negative affects (Cronbach $\alpha = 0.79$ and $0.73$ for the positive and negative aspects, respectively). Each item consists of an adjective-positive items: "Attentive, Active, Alert, Determined and Inspired"; and, negative items: "Hostile, Ashamed, Upset, Afraid and Nervous". Using a 7-point Likert scale, participants were asked to indicate to what extent each adjective corresponded to their emotional state when they were walking on the line.

### Physiological measures

The physiological measures (heart rate (HR) and breathing rate (BR)) were collected using the BioHarness thoracic belt (BH3; Zephyr Technology, Annapolis, USA). The electrocardiographic signal was sampled at 2 kHz through conductive silver fabric electrodes that are related to the transmitter. The HR (in beats per minute) was automatically calculated by internal algorithms of the BioHarness. The BR (in cycles per minute) was measured at 25 Hz by monitoring thoracic expansion using a strain gauge sensor. It should be noted that the reliability of the BioHarness has been adequately demonstrated for such cardiac (*Kim et al., 2013*) and respiratory (*Hailstone & Kilding, 2011*) measurements. To allow comparisons between distinct individuals, the heart and breath rates measured in each second during the tasks were normalized to their respective average during the last 30 s of rest period preceding that given task condition. Then, the rate variation for subject *i*, during task *T* and time instant *t* is given by Eq. (1):

$$\Delta R_{iT}(t) \ = \ \frac{R_{iT}(t) - R_{iT}}{R_{iT}} \tag{1}$$

in which $R_{iT}$ is the average of the rates measured at the last 30 s of the resting period.

## Electroencephalography measures

### EEG system and data preprocessing

Cortical activity was recorded at 1 kHz using 62 electrodes (mastoid electrodes were discarded), referenced at CPz, and positioned according to the 10-10 international electroencephalography (EEG) system. We used the eego[tm]sports (ANT Neuro, Enschede, The Netherlands) system, a portable equipment with a specific construction to reduce motion artefacts, enabling brain activity recording with moving subjects. The electrode impedance was initially maintained below 5 kΩ with the use of saline gel. The equipment's amplifier was stored in a large layer of cushioning in a small backpack.

For the first offline data analysis, the Advanced Source Analysis (ASA 4.9.3; ANT Neuro, Enschede, The Netherlands) software was used. Data were first re-referenced to the average of the 64 electrodes' signals. Next, the raw data were downsampled to 256 Hz

(since we were not interested in frequencies higher than the gamma band, and higher sampling frequencies would demand greater computational power, which in this case would be unnecessary) and bandpass-filtered between 0.5–50 Hz, with filter slope at 24 dB/oct using the Butterworth Zero phase Band-Pass filter, implemented in ASA 4.9.3. Afterwards, the data were exported to the eeg BRAIN Vision format, to be further processed using EEGLab, a MATLAB toolbox (*Delorme & Makeig, 2004*). The recorded time-series were visually inspected for motion artifacts, and segments which were clearly disrupted by movement (*e.g.*, which contained too many high-frequency and large-amplitude oscillations) were discarded: about 10–30% of data were excluded, depending on the subject; average recording time between subjects was of (8.1 ± 1.7) min, which accounts for (124,416 ± 26,112) data points in the EEG signals. Data points prior and post any discarded segments were concatenated to proceed with the preprocessing. Following manual signal exclusion, independent component analysis (ICA) (*Hyvärinen & Oja, 2000*) was used to separate the signal into 62 sources. Each independent component was then visually analyzed to identify the ones that contained mainly noise, which were excluded from the signals' reconstruction. Finally, data were band-pass filtered into four traditional frequency bands studied in the EEG literature: $\theta$ (4–8 Hz), $\alpha$ (8–13 Hz), $\beta$ (13–30 Hz) and $\gamma$ (30–50 Hz). For the electrodes' topography, the coordinates of the Talairach Atlas Space System (*Koessler et al., 2009*; *brmlab, 2011*) were considered.

### Estimates of the signal power—power spectral density

Since the band power of specific EEG frequencies relates to the task at hand (*Siegel, Donner & Engel, 2012*), one branch of our analysis consisted in studying how the power of distinct frequency bands was altered due to the high-walk and low-walk conditions.

The signals' power was computed for each electrode, condition and frequency band using the Welch's transform (*Welch, 1975*). To enable comparisons between individuals, power values during the tasks were normalized to their average during the rest period preceding that given task, in the same way as done for HR and BR. In other words, the 30 s rest immediately prior to either the high or low-walk was considered as a baseline measurement for each subject, with the following power variations being computed in relation to it for every 1s-segment the task was being performed. Mathematically, then, the power variation in electrode $e$ during task $T$, for each frequency band, subject and time instant $t$ $\left[\Delta P_{e,T}\right]$ can be written as:

$$\Delta P_{e,T} = \frac{P_{e,T}(t) - P_{e,T}}{P_{eT,}} \tag{2}$$

In Eq. (2), $P_{e,T}$ denotes the median power at the baseline state for that specific subject, frequency band, electrode and task. Note that, when computed in this manner, $\Delta P_{e,T}(t)$ actually represents relative variations to rest, with positive (negative) values corresponding to power increases (decreases).

### Functional connectivity analysis

Functional connectivity assumes that there may be a relation between the activity of two distinct brain areas, even when there is no anatomic linkage between them

(*Rogers et al., 2007*). A large variety of measures have been proposed to estimate FC in EEG signals (*Bastos & Schoffelen, 2016*). In this work, we opted for the coherence, since it is amongst the most commonly employed methodologies (*Bowyer, 2016*) in the field. Basically, coherence measures the synchronicity between the activity of two brain regions, mainly considering their phase difference. More specifically, even if two signals are out of phase from each other, they can still present a high coherence, should their corresponding phase difference remain approximately constant (*Srinivasan et al., 2007*). Amplitude is also accounted for, since two out-of-phase signals will tend to cancel out, and *vice versa* (*Siegel, Donner & Engel, 2012*).

Coherence matrices $A(t) = \{a_{ij}(t)\}_{62 \times 62}$ can be built, with each element corresponding to the coherence value between electrodes $i$ and $j$ at time $t$. Such matrices can be estimated for each subject and frequency band, for each 1s-segment of the tasks' performance. Seeking a reliable representation of the FC pattern for each condition, the matrices were time-averaged, yielding average connectivity matrices $\bar{A} = \{\bar{a}_{ij}\}$. These matrices were further binarized according to a threshold $r$ by setting:

$$b(r)_{ij} = \begin{cases} 1, & if \quad a_{ij} \geq r \\ 0, & otherwise \end{cases}.$$ (3)

Hence, the result is a set of binary matrices $B(r) = \{b(r)_{ij}\}_{62 \times 62}$ that can be analyzed for each $r$ value. In such a scenario, for each subject, $r$ relates to how often a given connection was present from the available samples under analysis. In our work, we set $r = 0.70$.

Following the calculations for the adjacency matrices, we assessed the resulting functional networks' topology through two widely employed graph metrics: the nodes' degree and clustering coefficient.

The degree for node $i$, $k_i$, represents the total number of connections it makes (*Fornito, Zalesky & Bullmore, 2016*):

$$k_i = \sum_{j=1}^{N} b_{ij}.$$ (4)

Thus, a node with high degree makes a large number of connections, and *vice-versa*.

On the other hand, the clustering coefficient for the $i^{th}$ node ($CC_i$) allows complementing this information by analyzing the connections among its nearest-neighbors. It can be mathematically expressed as (*Fornito, Zalesky & Bullmore, 2016*):

$$CC_i = \frac{2 \sum_{j=1}^{N} \sum_{l=1}^{N} b_{ij} b_{jl} b_{il}}{k_i(k_i - 1)}.$$ (5)

This measure then expresses local connectivity between nearest-neighbors.

Relative metrics' variations were investigated by analyzing how they varied in relation to their average resting values, similarly to the analysis done for the PSD based on Eq. (2).

## Statistical analysis

Data for psychological measures consisted of two answers (one for each task condition) per individual. The Wilcoxon test (*Gehan, 1965*), a nonparametric test for paired data, was used.

For physiological and EEG measures, the last 30 s of the rest period before the walking condition was considered as baseline for the statistical analysis. Mixed models were used to account for the non-independence of the repeated measures grouped within participants, an approach highly recommended for repeated measurements analysis (*Baayen, Davidson & Bates, 2008*). The mixed model included fixed factors representing the tasks (low- or high-walk), the order in which they were performed (first low- or first high-walk), and an interaction term between the tasks' factor and the order of the tasks, when statistically significant. As random effects, we included a random intercept and a random slope for the task effect, that could vary for each participant. Fixed effects were considered significant at a global alpha level of 0.05 and Bonferroni method for multiple comparisons was applied to correct the *p*-values. The goodness of fit of the models was assessed by the residuals' analysis of the normalized quantile residuals (*Dunn & Smyth, 1996*) *via* visual inspection of the quantile-quantile plot. Moreover, we considered a t-distribution for our models, which performed better than a classical normal mixed model.

All statistical analyses were performed in R (*R Core Team, 2019*), and the models were adjusted by the R package gamlss (*Rigby & Stasinopoulos, 2005*).

## Mixed models

To consider both random and fixed effects, our designed linear mixed models sought significant differences in HR, BR, PSD and FC, between the high- and low-walk conditions. This approach also enabled us to test whether the tasks performance's order was relevant.

Considering a response variable $Y_{ijt}$ representing a measure ($\Delta$HR, $\Delta$BR, $\Delta P$, $\Delta k_i$ or $\Delta CC_i$) of subject $i$ ($i = 1, 2, \ldots, 8$), performing task $j$ (low- or high-walk) at time $t$ ($t = 1, 2, \ldots, T_{ij}$); $T_{ij}$ represents the total time participant $i$ took to perform task $j$. We assumed $Y_{ijt}$ as having a t-Student distribution, with the model given by the following equation:

$$Y_{ijt} = (b_{0i} + \beta_0) + \beta_1 X_{1i} + (b_{2i} + \beta_2)X_{2i} + \beta_3 X_{1i}X_{2i} + \epsilon_{ijt}. \qquad (6)$$

In Eq. (6), $X_{1i}$ indicates the order in which subject $i$ performed the tasks, being equal to 1, if the high-walk was performed first, and 0, otherwise. $X_{2i}$ represents the task under analysis, being equal to one, if high-walk is currently under analysis, and 0, otherwise. $\beta_0$ is an intercept effect; $\beta_1$ represents the effect of tasks' order; $\beta_2$ represents a task effect; $\beta_3$ indicates an interaction effect between the tasks performance's order and the task being currently performed; $b_{0i}$ and $b_{2i}$ are random variables related to intra-subject variability, associated to the intercept term and to the slope of the task currently being performed, respectively. We assumed that $b_{0i} \sim N(0, \sigma_0^2)$ and $b_{2i} \sim N(0, \sigma_2^2)$, with $\sigma_0^2$ and $\sigma_2^2$

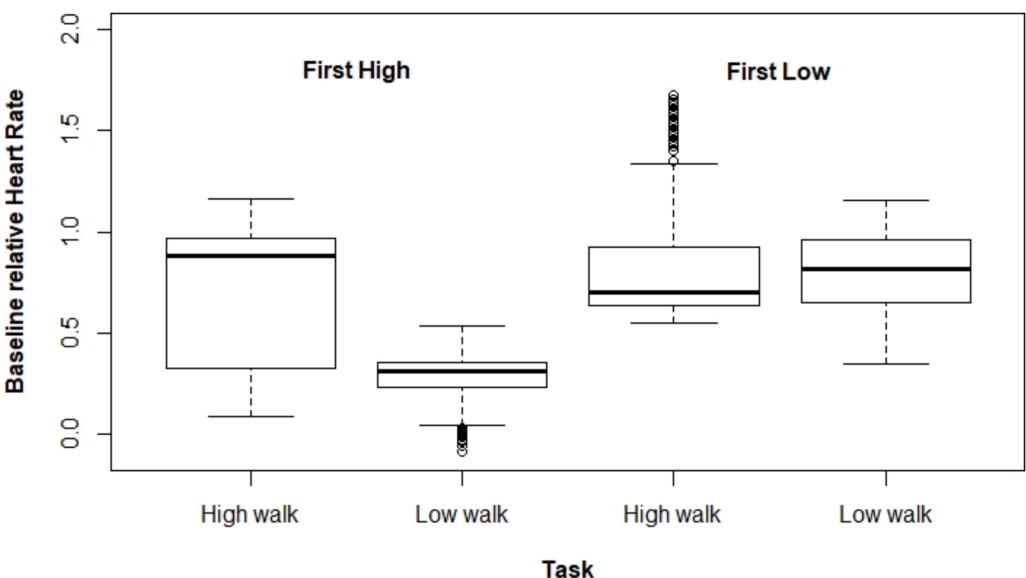

**Figure 2 Boxplot for ΔHR according to the tasks and their order of execution.**

representing variances across individuals. Finally, $\varepsilon_{ijt}$ is a random error variable, with $\epsilon_{ijt} \sim N(0, \ \sigma^2)$.

## RESULTS

In the following, the acronyms HW and LW will be used throughout the text to indicate high-walk and low-walk conditions, respectively.

### Psychological parameters

Concerning subjective flow experiences, HW presented $\Delta = 4.33$ (61.9% of the total possible score) on the flow FSS-2 scale. LW showed a higher score on the FSS-2, $\Delta = 5.56$ (79.43% of the total possible score), displaying a significant effect in relation to HW (Wilcoxon test, $p = 0.014$, mean difference = 1.656). For the concentration of attention and for the lack of effort to maintain attention (effortless), the Wilcoxon test did not reveal significant condition effects ($p = 0.072$ and $p = 0.203$, respectively). With respect to the Short Form of PANAS, the mean of the positive aspects did not show significant differences between conditions (Wilcoxon test, $p = 0.527$). For the mean of the negative emotions experienced in the task, HW displayed a higher score in relation to LW (mean difference = 2.05; Wilcoxon test, $p = 0.008$).

### Physiological parameters

Figures 2 and 3 present ΔHR and ΔBR boxplots for the measures of each subject according to the tasks and their executed order, respectively.

Regarding ΔHR, we observed a larger difference between the tasks when HW was performed first. The mixed model for ΔHR presented a significant interaction effect between the performed task and the order the tasks were executed ($p < 0.001$). The average ΔHR was higher at HW; however, the ΔHR difference between both tasks varied drastically

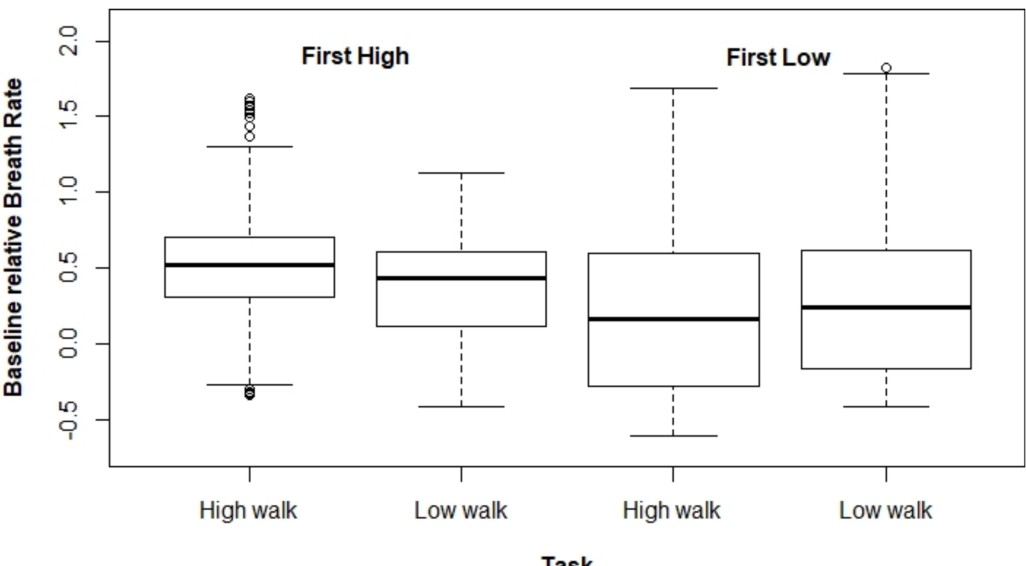

**Figure 3 Boxplot for ΔBR according to the task and the order the tasks were performed.**

**Table 1 Mean estimates for ΔHR according to the tasks order and the performed task obtained from the fitted mixed model.**

| Order | Task | Estimate | SE | CI 95% |
|---|---|---|---|---|
| HW first | HW | 0.736 | 0.002 | [0.731–0.741] |
| | LW | 0.272 | 0.002 | [0.268–0.276] |
| LW first | HW | 0.865 | 0.003 | [0.860–0.870] |
| | LW | 0.820 | 0.003 | [0.815–0.826] |

Note:
HW, high-walk; LW, low-walk; SE, standard error; CI, confidence interval.

**Table 2 Estimates of mean ΔBR according to the order and the task performed obtained from the fitted mixed model.**

| Order | Task | Estimate | SE | CI 95% |
|---|---|---|---|---|
| HW first | HW | 0.522 | 0.010 | [0.502–0.541] |
| | LW | 0.367 | 0.010 | [0.348–0.386] |
| LW first | HW | 0.229 | 0.012 | [0.205–0.252] |
| | LW | 0.273 | 0.014 | [0.246–0.301] |

Note:
HW, high-walk; LW, low-walk; SE, standard error; CI, confidence interval.

depending on which type of walking was performed first, being 0.464 for HW executed first, and 0.045 for the other case (Table 1).

The mixed model adjusted for ΔBR also presented a significant interaction effect between the task performed and the tasks order ($p < 0.001$). In the same way as for ΔHR, ΔBR was higher at HW when this task was carried out first (estimated mean difference = 0.155). However, for the other scenario (*i.e.*, performing LW first), there was
no significant difference between both tasks (estimated mean difference = –0.044). Moreover, we observed that ΔBR was larger when HW was performed first (Table 2).

## Neural activity results

### *Power spectral density*

Since the order in which the tasks were performed was relevant, mean $\Delta P$ estimates, obtained through the model in Eq. (6), were compared across the two walking situations. Subjects were divided in two groups–those who performed HW first and those who performed LW first–, which were analyzed separately. In each case, average $\Delta P$ values were subtracted between HW and LW. These results are displayed in Fig. 4, in which plotted values correspond to $(\Delta P_{HW} - \Delta P_{LW})$. The top row in Figs. 4A and 4B shows results for performing HW first, whereas the bottom row indicates the other situation. Frequency bands ($\theta$, $\alpha$, $\beta$, $\gamma$) are displayed in columns. Visualizing results in this manner enables enhancing differences across the walking conditions at the group level while considering the tasks' order. Thus, in Fig. 4A, red (positive values) tones indicate $\Delta P_{HW} > \Delta P_{LW}$, and blue (negative values) tones indicate $\Delta P_{HW} < \Delta P_{LW}$. In other words, a mostly blue plot indicates predominance of larger $\Delta P_{LW}$ values, and *vice versa*.

Figure 4B displays electrodes whose $(\Delta P_{HW} - \Delta P_{LW})$ were statistically significant. Red (blue) circles indicate significantly larger (smaller) $\Delta P_{HW}$ values. Smooth colored regions under electrodes' locations represent brain functional networks: Primary Somato-Sensory Cortex (white region); Default Mode Network-DMN (orange region) (*Rojas et al., 2018*); Ventral Visual Stream Network (grey regions); Dorsal Visua. Stream Network (purple region) (*Goodale & Milner, 1992*; *Milner & Goodale, 2008*; *Norman, 2002*); Ventral Attention Network–VAN (yellow regions); Dorsal Attention Network-DAN (soft blue regions); and Fronto Parietal Control Network–FPN (soft blue + pink regions) (*Japee et al., 2015*). Regarding our higher flow scores, *i.e.*, in LW, when both attention networks VAN and DAN, displayed significant $\Delta P$ simultaneously at the same brain hemisphere close areas (*i.e.*, yellow and soft blue 'sub-networks' at frontal or parietal-temporal regions), it were marked as 'collaborating regionally' (CR), or, conversely, when electrodes related to only one, VAN or DAN, presented higher significant $\Delta P$, the regions were marked as 'prevailing regionally' (PR). It can be seen that regardless of condition order performance, when VAN-DAN CR was not the case, hemispheric VAN PR appeared at frontal regions in all bands but $\alpha$ for the right when HW was first, and $\beta$ for the left when LW was the case, evidencing left lateralization tendency. There was no DAN PR at frontal areas during LW (Fig. 4B) top and bottom rows. Differently, the behavior of VAN and DAN PR at the parietal-temporal regions were more dependent on which condition was performed first. VAN PR was found only when LW was first. DAN PR was found in the two slower frequency bands, $\theta$ and $\alpha$, tending to the right hemisphere. Considering yet, the most intense significant $\Delta P$ in the faster frequency bands, $\beta$ and $\gamma$ when LW was first (deeper intense blue in Fig. 4A, bottom row), there was no clear consistent VAN or DAN overall PR at parietal-temporal areas in LW.

From Fig. 4, it can be seen that some areas changed according to the condition carried out first, despite whether other brain regions remained approximately the same, wherever

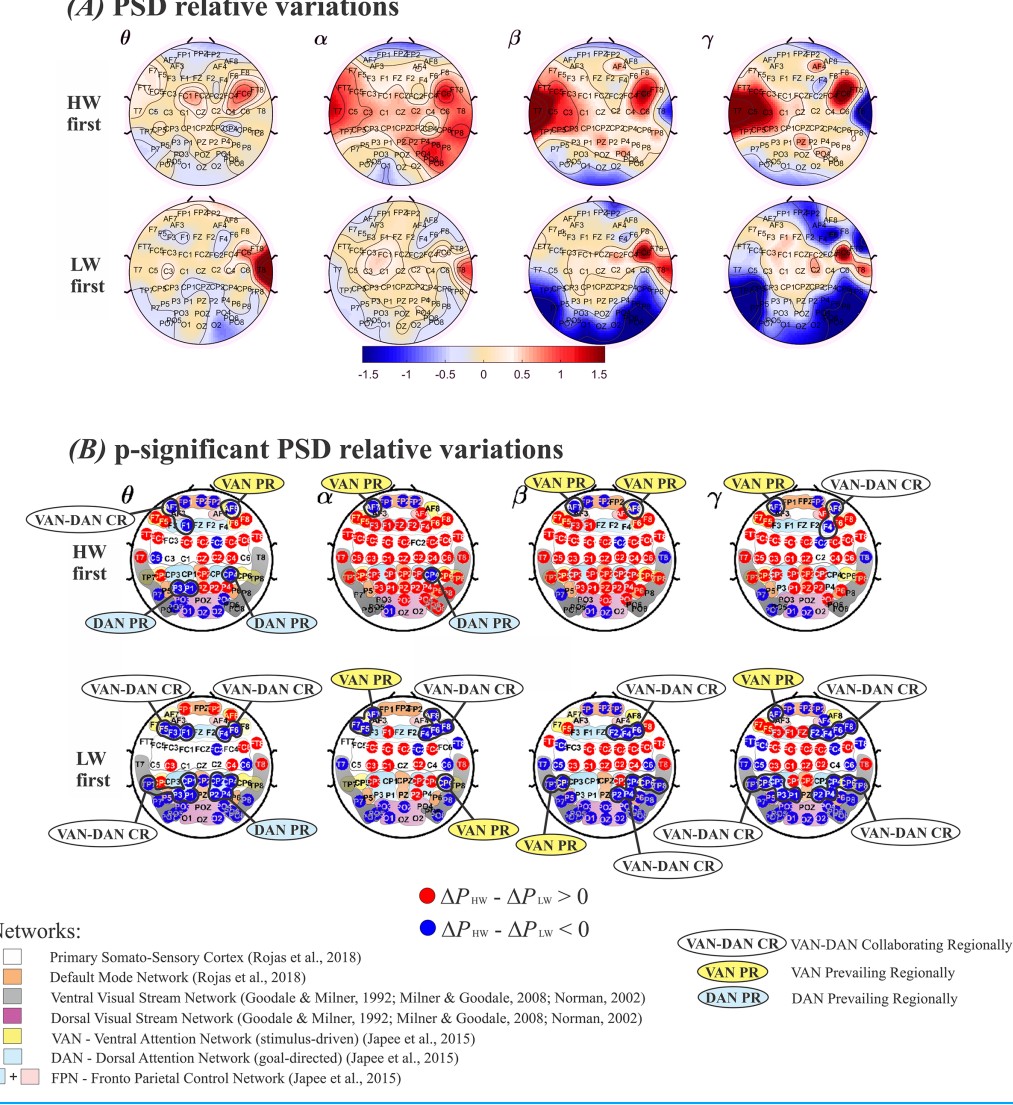

**(A) PSD relative variations**

**(B) p-significant PSD relative variations**

Networks:

☐ Primary Somato-Sensory Cortex (Rojas et al., 2018)
🟧 Default Mode Network (Rojas et al., 2018)
⬜ Ventral Visual Stream Network (Goodale & Milner, 1992; Milner & Goodale, 2008; Norman, 2002)
🟪 Dorsal Visual Stream Network (Goodale & Milner, 1992; Milner & Goodale, 2008; Norman, 2002)
🟨 VAN - Ventral Attention Network (stimulus-driven) (Japee et al., 2015)
🟦 DAN - Dorsal Attention Network (goal-directed) (Japee et al., 2015)
🟦 + 🟥 FPN - Fronto Parietal Control Network (Japee et al., 2015)

🔴 $\Delta P_{HW} - \Delta P_{LW} > 0$
🔵 $\Delta P_{HW} - \Delta P_{LW} < 0$

VAN-DAN CR  VAN-DAN Collaborating Regionally
VAN PR  VAN Prevailing Regionally
DAN PR  DAN Prevailing Regionally

**Figure 4 Plots of $(\Delta P_{HW} - \Delta P_{LW})$ according to the model in Eq. (6).** (A) Positive values represent a larger relative power ($\Delta P$) during HW, and *vice-versa*. Top row: HW executed first; bottom row: LW executed first. (B) Electrodes whose ($\Delta P_{HW} - \Delta P_{LW}$) differences were statistically significant. Top row: HW executed first; bottom row: LW executed first. The color scheme follows the setup shown in Fig. 4 (A): red (blue) circles indicate HW (LW) statistically meaningful effects. The corresponding functional network more likely to encompass each electrode is also shown: Primary Somato-Sensory Cortex (white region); DMN (orange region) (*Rojas et al., 2018*) ; Ventral Visual Stream Network (grey regions); Dorsal Visual Stream Network (purple region) (*Goodale & Milner, 1992*; *Norman, 2002*; *Milner & Goodale, 2008*); VAN (yellow regions); DAN (soft blue regions); FPN (soft blue + pink regions) (*Japee et al., 2015*). HW, high-walk; LW, low-walk. Regarding LW, the same brain hemispheric significant $\Delta P$ simultaneously at frontal or parietal-temporal lobes within VAN and DAN, are marked as 'collaborating regionally' (CR). Electrodes related to only one attention network, presenting higher significant $\Delta P$ on these specific close areas, are marked as VAN or DAN 'prevailing regionally' (PR).

HW or LW presented significantly larger $\Delta P$ effect regardless of which task was executed first. In the Supplemental Material, see Table 3, presenting the functional networks according to references (*Goodale & Milner, 1992*; *Japee et al., 2015*;

**Table 3 Brain networks and corresponding (electrodes) that presented significant differences between PSD variations (ΔP) for HW and LW, according to the condition executed first and to the frequency band.**

| Order | Freq. band | Effect | |
|---|---|---|---|
| | | $\Delta P_{HW} - P_{LW} > 0$ | $\Delta P_{HW} - PLW < 0$ |
| HW first | $\theta$ | PSSC (FCz, FC1, FC4, FC5, FC6*, Cz, C2, C4*) | PSSC (FC2*, C5) |
| | | DMN (CPz, Pz) | DMN (FPz, FP1, FP2) |
| | | VVSN (T7) | VVSN (P7*, PO5*) |
| | | VAN (F5, F6, F7, F8, CP5*) | DVSN (PO3*, PO4*, Oz, O1, O2*) |
| | | FT (FT7, FT8*) | VAN (AF7, AF8) |
| | | | DAN (F1*, CP4*, P1*, P3*) |
| | $\alpha$ | PSSC (FCz*, FC1*, FC3*, FC4*, FC5, FC6, Cz, C1*, C2*, C3*, C4, C5, C6) | DMN (FPz, FP1, FP2) |
| | | DMN (CPz, Pz, P6) | DVSN (O1*) |
| | | VVSN (T7, T8*, TP7, TP8, P8, PO6, PO8) | VAN (AF7*) |
| | | DVSN (POz, PO4) | DAN (CP4) |
| | | VAN (F5, F6, F7, F8, CP5, CP6, TP7, TP8) | |
| | | DAN (Fz, F1*, F2, F3, F4, CP1, CP2*, CP3*, P1, P2*, P3, P4) | |
| | | FPN (same as DAN + AF4) | |
| | | FT (FT7, FT8) | |
| | $\beta$ | PSSC (FCz*, FC1*, FC3, FC4*, FC5, FC6*, Cz*, C1*, C2*, C3*, C4*, C5, C6) | PSSC (FC2) |
| | | DMN (CPz, Pz, P6) | DMN (FPz*, FP1*) |
| | | VVSN (T7, TP7, TP8) | VVSN (T8, P7*) |
| | | DVSN (POz, PO4) | DVSN (Oz*, O1*) |
| | | VAN (F5*, F6, F7, F8*, CP5, CP6, TP7, TP8) | VAN (AF7, AF8) |
| | | DAN (F1, F3, CP1, CP2*, CP3, CP4, P1, P2, P3, P4) | |
| | | FPN (same as DAN + AF4) | |
| | | FT (FT7*, FT8*) | |
| | $\gamma$ | PSSC (FCz*, FC1*, FC3*, FC4, FC5, FC6*, Cz*, C1*, C3*, C4, C5, C6) | PSSC (FC2) |
| | | DMN (CPz, Pz, P6) | DMN (FP2*) |
| | | VVSN (T7, TP7, TP8) | VVSN (T8, P7*, PO7*) |
| | | DVSN (PO4) | DVSN (Oz*, O1*) |
| | | VAN (F5*, F6, F7*, F8, CP5, TP7, TP8) | VAN (AF7, AF8) |
| | | DAN (CP1*, CP2, CP3, P1, P2) | DAN (F4*) |
| | | FPN (same as DAN + AF4) | |
| | | FT (FT7, FT8) | |
| LW first | $\theta$ | PSSC (FC6*, C3, C4*) | PSSC (FC2*, C6) |
| | | DMN (FP1, FP2) | DMN (P5) |
| | | VVSN (T8) | VVSN (TP7, P7*, P8, PO5*, PO6, PO7, PO8) |
| | | VAN (AF8, CP5*) | DVSN (PO3*, PO4*, O2*) |
| | | FT (FT8*) | VAN (F5, F6, TP7) |
| | | | DAN (F1*, F3, F4, CPz, CP1, CP2, CP4*, P1*, P2, P3*, P4) |

| | | Effect | |
|---|---|---|---|
| **Order** | **Freq. band** | **$\Delta P_{HW} - P_{LW} > 0$** | **$\Delta P_{HW} - PLW < 0$** |
| | α | PSSC (FCz*, FC1*, FC3*, FC4*, C1*, C2*, C3*) | PSSC (FC2, C6) |
| | | VVSN (T8*) | VVSN (T7, P7, P8, PO5, PO7, PO8) |
| | | DAN (F1, CP2*, CP3*, CP4, P2*) | DVSN (POz, PO3, O1*) |
| | | | VAN (AF7*, AF8, F5, F6, F7, F8, CP6) |
| | | | DAN (F4); FT (FT8) |
| | β | PSSC (FCz*, FC1*, FC2, FC4*, FC6*, Cz*, C1*, C2*, C3*, C4*) | PSSC (FC5, C5, C6) |
| | | VVSN (T8) | DMN (FPz*, FP1*, FP2, P5, P6) |
| | | VAN (F5*, F8*) | VVSN (T7, TP7, TP8, P7*, P8, PO5, PO6, PO7, PO8) |
| | | DAN (CP2*) | DVSN (POz, PO3, PO4, Oz*, O1*, O2) |
| | | FT (FT7*, FT8*) | VAN (F6, CP5, CP6, TP7, TP8) |
| | | | DAN (F2, F4, CP4, P2, P4) |
| | γ | PSSC (FCz*, FC1*, FC2, FC3*, FC6*, Cz*, C1*, C2, C3*) | PSSC (FC4, FC5, C5, C6) |
| | | VVSN (T8) | DMN (FPz, FP1, FP2*, P5, P6) |
| | | VAN (F5*, F7*) | VVSN (T7, TP7, TP8, P7*, P8, PO5, PO6, PO7*, PO8) |
| | | DAN (Fz, F1, F3, CP1*) | DVSN (PO3, PO4, Oz*, O1*, O2) |
| | | | VAN (F6, F8, CP5, CP6, TP7, TP8) |
| | | | DAN (F2, F4*, CP3, CP4, P1, P2, P3, P4) |
| | | | FPN (same as DAN + AF4) |
| | | | FT (FT7, FT8) |

**Note:**
HW, high-walk; LW, low-walk; PSSC, primary somato-sensory cortex; DMN, default mode network; VVSN, ventral visual stream network; DVSN, dorsal visual stream network; VAN, ventral attention network; DAN, dorsal attention network; FPN, fronto parietal control network; FT, fronto temporal regions. Electrodes presenting significantly larger ($\Delta P$) effect in the same frequency band regardless task order are marked by *.

Milner & Goodale, 2008; Norman, 2002; Rojas et al., 2018), and corresponding electrodes which showed significant differences among conditions.

### Functional connectivity

Results for the degree and the clustering coefficient (CC), are shown in Figs. 5A and 5B, respectively. As in Fig. 4, the top row shows results for performing HW first, whereas the bottom row indicates LW first. Each frequency band (θ, α, β, γ) is displayed as columns. Electrodes with statistically significant ($\Delta k_{i\,HW} - \Delta k_{i\,LW}$) or ($\Delta CC_{i\,HW} - \Delta CC_{i\,LW}$) values are plotted as: red circles, for larger $\Delta k_{i\,HW}$ or $\Delta CC_{i\,HW}$; and as blue circles, for larger $\Delta k_{i\,LW}$ or $\Delta CC_{i\,LW}$. The radius' size was normalized between HW and LW and represents the intensity of significance between the two conditions. Under the electrodes' positions, smooth colored regions refer to the same brain functional networks as before (see Fig. 4).

Throughout frequency bands, regardless of which condition was performed first, Figs. 5A and 5B show greater significant electrode areas at most networks for HW in both FC graph metrics, Degree and CC, comparing to LW, as well as, overall larger intensity at the significant nodes (normalized red radius sizes in relation to blue ones).

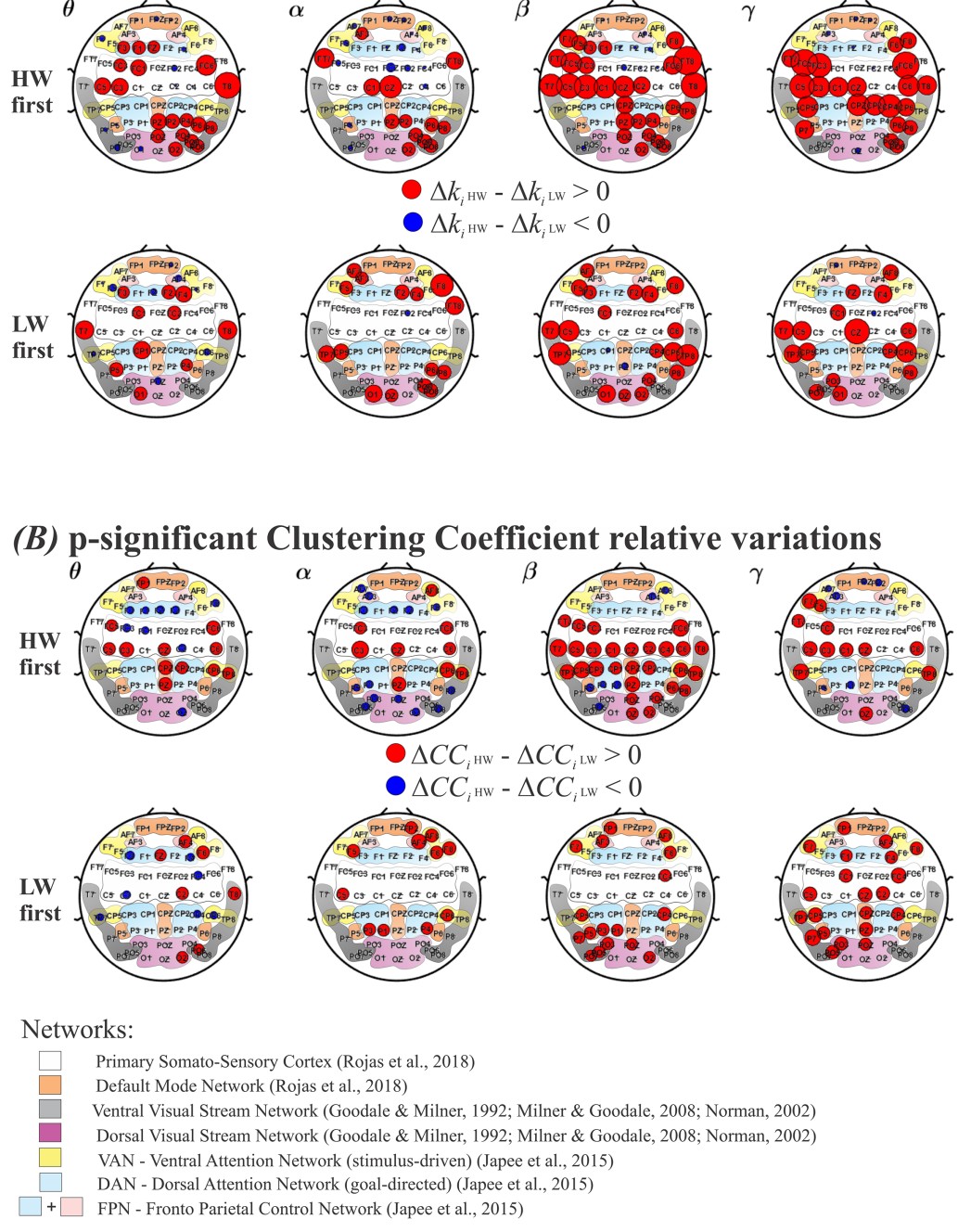

**Figure 5 Variations for the graph metrics.** (A) Electrodes whose degree variation differences ($\Delta K_{i\ HW} - \Delta K_{i\ LW}$) were statistically significant according to the model in Eq. (6). Top row: HW executed first; bottom row: LW executed first. (B) Electrodes whose clustering coefficient variation differences ($\Delta CC_{i\ HW} - \Delta CC_{i\ LW}$) were statistically significant according to the model in Eq. (6). Top row: HW executed first; bottom row: LW executed first. Red (blue) circles indicate HW (LW) statistically meaningful effects. The radius size represents the intensity of change in each metric. Corresponding functional networks are shown in the same scheme defined in Fig. 4. HW, high-walk; LW, low-walk.

**Table 4 Brain networks and corresponding (electrodes) that presented significant differences between degree variations ($\Delta k_i$) for HW and LW, according to the condition executed first and to the frequency band.**

| Order | Freq. band | Effect | |
| --- | --- | --- | --- |
| | | $\Delta k_{i\,HW} - \Delta k_{i\,LW} > 0$ | $\Delta k_{i\,HW} - \Delta k_{i\,LW} < 0$ |
| HW first | θ | PSSC (FC1*, FC3, FC6, C3, C5) | PSSC (FC2, C2) |
| | | DMN (Pz, P6) | DMN (FPz, FP1, FP2, P5) |
| | | VVSN (T8*, P8, PO6, PO8) | VVSN (P7, PO7) |
| | | DVSN (POz, PO4, O2) | DVSN (O1) |
| | | DAN (Fz, F1, F3*, P2, P4*) | VAN (F7*) |
| | | | DAN (F4) |
| | α | PSSC (Cz, C1) | PSSC (FCz, FC2*, FC4, FC5, C4) |
| | | DMN (Pz) | DMN (FPz, FP2, P5) |
| | | VVSN (P8*, PO6*, PO8*) | VVSN (PO7) |
| | | DVSN (PO4, O2) | VAN (AF7, AF8, F5) |
| | | DAN (P2) | DAN (Fz, F2, F4) |
| | | FPN (same as DAN+AF3*) | FPN (same as DAN+AF4) |
| | | FT (FT7, FT8*) | |
| | β | PSSC (FC3, FC5, FC6, Cz, C1, C2, C3, C5*) | PSSC (FCz, FC2, FC4) |
| | | DMN (CPz, Pz) | DMN (FP1); VVSN (PO7) |
| | | VVSN (T7*, T8, TP8*, PO6, PO8) | DAN (Fz, F2, F4) |
| | | DVSN (POz, PO4*, O2*) | FPN (same as DAN+AF4) |
| | | VAN (F5*, F7, F8*, CP6*, TP8) | |
| | | DAN (F1, F3*, P2, P4) | |
| | | FT (FT7, FT8) | |
| | γ | PSSC (FC3, FC5, FC6, Cz*, C1, C3, C5*, C6*) | PSSC (FC2*, FC4) |
| | | DMN (CPz, P6) | DMN (FPz, FP1*, FP2) |
| | | VVSN (T8, P7, P8*, PO6, PO8) | DVSN (Oz) |
| | | VAN (F6, F8, CP5*, CP6*) | DAN (Fz) |
| | | DAN (CP2, CP4*) | FPN (same as DAN+AF3) |
| | | FT (FT7) | |
| LW first | θ | PSSC (FC1*, FC2) | DMN (FP2) |
| | | DMN (P5); VVSN (T7, T8*) | VVSN (TP7) |
| | | DVSN (O1) | DVSN (POz) |
| | | DAN (F2, F3*, F4, CP1, P4*) | VAN (F5, F6, F7*, CP6, TP7) |
| | | | DAN (Fz) |
| | | | FPN (same as DAN+AF4) |
| | α | DMN (P6) | PSSC (FC2*) |
| | | VVSN (TP7, P8*, PO6*, PO8*) | |
| | | DVSN (OZ, O1) | |
| | | VAN (F5, F8, CP5,TP7) | |
| | | DAN (F2, F4) | |
| | | FPN (same as DAN+AF3*) | |
| | | FT (FT8*) | |

(Continued)

| | | Effect | |
|---|---|---|---|
| Order | Freq. band | $\Delta k_{i\,HW} - \Delta k_{i\,LW} > 0$ | $\Delta k_{i\,HW} - \Delta k_{i\,LW} < 0$ |
| | β | PSSC (FC1, C5*, C6) | DMN (Pz) |
| | | VVSN (T7*, TP7, TP8*, P8) | DAN (CP1) |
| | | DVSN (PO4*, Oz, O1, O2*) | |
| | | VAN (AF7, F5*, F8*, CP6*, TP7, TP8) | |
| | | DAN (F2, F3*, F4, CP4) | |
| | γ | PSSC (FC1, Cz*, C5*, C6*) | PSSC (FC2*) |
| | | VVSN (T7, TP7, P8*, PO7) | DMN (FP1*) |
| | | DVSN (PO3, O1) | |
| | | VAN (AF8, CP5*, CP6*, TP7) | |
| | | DAN (Fz, F4, CP4*) | |

Note:
HW, high-walk; LW, low-walk; PSSC, primary somato-sensory cortex; DMN, default mode network; VVSN, ventral visual stream network; DVSN, dorsal visual stream network; VAN, ventral attention network; DAN, dorsal attention network; FPN, fronto parietal control network; FT, fronto temporal regions. Electrodes presenting significantly larger ($\Delta k_i$) effect in the same frequency band regardless task order are marked by *.

**Table 5 Brain networks and corresponding (electrodes) that presented significant differences between clustering coefficient variations ($\Delta CC_i$) for HW and LW, according to the condition executed first and to the frequency band.**

| | | Effect | |
|---|---|---|---|
| Order | Freq. band | $\Delta CC_{i\,HW} - \Delta CC_{i\,LW} > 0$ | $\Delta CC_{i\,HW} - \Delta CC_{i\,LW} < 0$ |
| HW first | θ | PSSC (FC5, FC6, Cz, C3, C5, C6) | PSSC (FC1, FC3, C2) |
| | | DMN (FP1, CPz, Pz) | DVSN (O2) |
| | | VVSN (TP8) | VAN (F8) |
| | | VAN (CP6, TP8) | DAN (Fz, F1, F2, F3*) |
| | | DAN (CP2) | |
| | α | PSSC (FC3, FC6, Cz, C3, C6) | DMN (P5) |
| | | DMN (Pz) | VVSN (P8, PO6, PO8) |
| | | VVSN (TP8) | DVSN (POz, PO3, O2) |
| | | VAN (AF8*, CP6*, TP8) | VAN (AF7, F6) |
| | | | DAN (Fz, F2, F3, P3) |
| | | | FPN (same as DAN+AF3) |
| | β | PSSC (FC3, FC5, FC6, Cz, C1, C2, C3, C4, C5, C6) | DMN (P5) |
| | | DMN (CPz, Pz, P6) | VVSN (PO5) |
| | | VVSN (T7, T8, P7*, P8, TP8, P8) | VAN (AF8) |
| | | DVSN (POz*, PO4, Oz, O2*) | DAN (P1, P3) |
| | | VAN (CP5*, CP6, TP7, TP8) | FPN (same as DAN+AF4) |
| | | DAN (CP1, CP2, CP3) | |
| | | FT (FT7) | |

| | | Effect | |
|---|---|---|---|
| Order | Freq. band | $\Delta CC_{i\,HW} - \Delta CC_{i\,LW} > 0$ | $\Delta CC_{i\,HW} - \Delta CC_{i\,LW} < 0$ |
| | $\gamma$ | PSSC (FC3, Cz*, C1, C3, C5*, C6) | DMN (FPz, FP2, P5) |
| | | VVSN (TP7*, TP8) | VVSN (PO8) |
| | | VAN (F5, F7*, TP7*, TP8) | VAN (AF7) |
| | | FT (FT7) | DAN (P1) |
| | | | FPN (same as DAN+AF3) |
| LW first | $\theta$ | PSSC (C2) | PSSC (FC4, C3) |
| | | VVSN (T8, PO6) | VVSN (TP7) |
| | | DVSN (O2) | VAN (CP6, TP7) |
| | | VAN (F6) | DAN (F3*, F4, CP4) |
| | | DAN (Fz) | |
| | | FPN (same as DAN+AF4) | |
| | $\alpha$ | PSSC (C5) | |
| | | DMN (FP2) | |
| | | VAN (AF8*, F5, F6, F8, CP6*) | |
| | | DAN (P1, P3, P4) | |
| | | FPN (same as DAN+AF4) | |
| | $\beta$ | PSSC (FC4) | |
| | | DMN (FP1, P5) | |
| | | VVSN (P7*, PO5, PO7) | |
| | | DVSN (POz*, PO3, O2*) | |
| | | VAN (AF8, F6, F7, CP5*) | |
| | | DAN (CP4, P1, P3) | |
| | | FPN (same as DAN+AF3) | |
| | $\gamma$ | PSSC (FC1, FC4, Cz*, C2, C5*) | |
| | | DMN (FP1, CPz, Pz, P5) | |
| | | VVSN (TP7*, P7, PO5) | |
| | | DVSN (POz, PO3) | |
| | | VAN (F7*, F8, CP5, TP7*) | |
| | | DAN (F1, F4, CP4) | |
| | | FPN (same as DAN+AF4) | |

**Note:**
HW, high-walk, LW, low-walk. PSSC, primary somato-sensory cortex; DMN, default mode network; VVSN, ventral visual stream network; DVSN, dorsal visual stream network; VAN, ventral attention network; DAN, dorsal attention network; FPN, fronto parietal control network; FT, fronto temporal regions. Electrodes presenting significantly larger ($\Delta CC_i$) effect in the same frequency band regardless task order are marked by *.

Because of the normalized radius size, for a better view, in the Supplemental Material, see Table 4 for the Degree, and Table 5 for the CC, presenting the functional networks according to references (*Goodale & Milner, 1992*; *Japee et al., 2015*; *Milner & Goodale, 2008*; *Norman, 2002*; *Rojas et al., 2018*), and corresponding electrodes which showed significant differences among conditions.

It is interesting to note that LW presented yet less significant regions in faster frequencies when LW was performed first. This can be seen for Degree, where LW showed significance only on FC2 in the α band; while on PZ and CP1 in the β band; and on FC2 and FP1 in the γ band; and it is even more evident for CC, where LW did not present any significant sensor in the α, β, or γ band frequencies.

## DISCUSSION

Psychological investigations have linked extreme sports with flow experiences (*Breton, 2000*; *Brymer & Schweitzer, 2017*; *Jackson & Csikszentmihalyi, 1999*; *Moen et al., 2017*; *Partington, Partington & Olivier, 2009*). Here we employed a slackline-based protocol with varying altitudes to promote a kind of 'relaxed subjective tension' during action, attempting to induce '(effortless) relaxed attention' (*Bruya, 2010*) to elicit flow. We examined the psychological, peripheral-physiological and neurophysiological responses associated with qualitative attention, flow experience, and positive/negative emotions during a close-to-real-world extreme sport situation. We did not find statistically significant differences of subjective attention and effortless attention between HW and LW. Highest scores on the FSS-2 flow scale questionnaire were higher for LW, whereas the negative emotion aspects were higher at HW. It has been suggested that 60% of the total possible score on FSS-2 can be considered as indicating some degree of flow experience (*Jackson & Eklund, 2004*; *Kawabata & Mallett, 2011*). HW and LW presented 61.9% and 79.43%, respectively, of the total possible score on FSS-2. Considering that participants did not hold large experience with HW, while the protocol does not configure a real risk situation (due to participants being attached to safety ropes), the results on FSS-2 suggest that the subjects' skills might have been balanced with the walking challenges to provide degrees of flow experiences, while the negative emotion aspects led to some inhibition of flow during HW.

These indications are reinforced by the higher ΔHR during HW, also suggesting that negative emotions, such as fear, tension, *etc.*, yielded sympathetic activation, especially when HW was carried out first. However, when studying LW-first results, ΔHR differs considerably less between the two conditions. Complementarily, ΔBR results were also higher in HW when performing HW first. Fast BR is a sympathetic domain signal; therefore, these results indicate a collaboration of this domain to inhibit flow experiences. For LW-first, we observed an inverse trend, with LW showing slightly higher, although not significant, ΔBR than that shown in HW.

Overall, results for ΔHR and ΔBR, when considering LW first, could suggest an equilibrium between sympathetic and parasympathetic tones during HW. Indeed, other studies have found sympathetic-parasympathetic equilibrium related to flow experiences (*de Manzano et al., 2010*; *Harmat et al., 2015*; *Peifer et al., 2014*; *Tozman et al., 2015*; *Ullén et al., 2010*). Based on several psychological reports (*e.g.*, a free solo climber (free of security ropes), compares flow as a hyper-focused and, at the same time, calming, experience (*Sparks, 2016*)), on these peripheral-physiological responses, and on some of our neurophysiological results, we suggest that scientific experiments using HW protocols, as well as other extreme sports, can potentially promote simultaneous sympathetic-parasympathetic engagement.

This might induce calming sensations when dealing with fear control in order to be able to cope with the task at hand, leading to a propensity of flow experiences.

It is highly conceivable that the EEG "flow signature" will be polyrhythmic and supported by different mechanisms (*Cheron, 2016*). Hence, to better comprehend the complex brain functioning behavior through different emotions and feelings, a set of theories and points-of-view should be considered beyond EEG literature. In the following, we discuss some theoretical approaches and how its relate to the brain measures under investigation, *i.e.*, relative differences in the signals' power and functional connectivity metrics.

Mainly, our $\Delta P$ results lead us to consider that the brain's signatures cannot be well generalized only across tasks and conditions, but must be understood as highly context-dependent, being altered by perception, cognition, emotion, motivation, action *etc.*, (*Pessoa, 2017*). The fact that we are calculating $\Delta P$ as relative-to-baseline values (Eq. (2)), and that the average $\Delta P$ values were subtracted between HW and LW, implies that the observed results reflect that the signal's power in the mentioned areas undergoes actual variation when compared to the baseline condition (rest), and when contrasted between the two altitudes. Our main findings raised three hypotheses:

**(H1) The order of condition performed first leads to different $\Delta P$ lateralization between brain hemispheres.** The context of the conditions, such as the task's order, influence psychological aspects, which reflect in $\Delta P$, such as changing lateralization increases between brain hemispheres. The hemispheric laterality valence hypothesis (*Hellige, 1993*) proposes dominance of the right hemisphere for processing negative, unpleasant emotions, and dominance of the left hemisphere for positive, pleasant ones. While our psychological negative emotion (PANAS) and peripheral physiological results presented signals of emotional arousal when HW was performed first, we had the left hemisphere with higher $\Delta P$ in a larger region across all frequency bands for HW. The $\Delta P$ results contradict the hemispheric laterality valence hypothesis when performing HW first. Relating to HW with LW first, our results could be considered as sustaining the lateralization hypothesis: HW showed right lateralized fronto-temporal higher $\Delta P$ through all frequency bands (Figs. 4A and 4B, bottom row), while it displayed a smaller area at the left hemisphere and central brain regions, compared to when starting with HW (Fig. 4B). But, interestingly, the peripheral physiology revealed a sympathetic/parasympathetic equilibrium in HW when LW was first, and HW presented significant $\Delta P$ at the vmPFC region, for both hemispheres, in the $\theta$ frequency (Fig. 4B bottom row, first column). vmPFC is related to counterbalancing amygdala activation during emotional arousal as in fear control (*Pessoa, 2017*, *2018*). These findings could be interpreted as a reflex of a mental strategy to handle negative emotions and promote calm conduction (*i.e.*, significantly higher $\Delta P$ at vmPFC only in the "slow-frequency" $\theta$ band and sympathetic-parasympathetic equilibrium). Concerning the hemispheric valence hypothesis and these contrary results, it could be argued that HW could also promote positive/pleasant emotions, competing with the negative. Managing and overcoming fear and anxiety can provide a medium to facilitate profound and positive transformations (*Holmbom, Brymer & Schweitzer, 2017*). We did not find significant results with the positive emotions on the PANAS questionnaire for any

condition, but LW was related to the experience of high flow, and flow is commonly associated with enjoyment emotions (*Pels, Kleinert & Mennigen, 2018*; *Rufi et al., 2016*). Our results could suggest a LW trend towards positive emotions linked to a slightly higher $\Delta P$ at occipital, parietal and temporal left hemisphere (Figs. 4A and 4B). However, LW also showed higher $\Delta P$ in the right hemisphere, markedly at the temporal lobe in the $\beta$ and $\gamma$ bands when HW was first. Some inconsistencies of our results can be derived from the tasks' performance order, which apparently influenced the involved emotions and results. Differences in $\Delta P$ between tasks were thought to be mainly due to a distinct visual stimulus of height. Previous EEG studies have reported habituation of visual responses at different stimulus conditions (*e.g.*, *Brust-Carmona et al., 2014*; *Cao et al., 2020*). This might indicate a type of training saturation mechanism in the brain for similar types of tasks conducted in a relatively close time interval. It could mean that performing either of the walking conditions first causes the $\Delta P$ response at the subsequent task to be damped. It can be perceived that larger brain areas are dominated by the performed first condition (Figs. 4A and 4B). Hence, at this point, our results support that the tasks' order cannot be disregarded in this type of study and, thus, similarly-designed experiments ought to consider this and other sorts of complexities in their hypotheses.

**(H2) Regardless of which condition is performed first, negative emotion aspects induce an increase in $\Delta P$ at the sensorimotor cortex and parietal medial areas during HW, while high scores on the flow questionnaire coincide with persistent higher $\Delta P$ at medial prefrontal, middle-frontal and occipital areas in LW.** HW, related to greater negative emotion aspects, presented higher $\Delta P$ at central brain areas, while larger flow experiences in LW correspond mainly to higher $\Delta P$ at medial prefrontal, middle-frontal and occipital cortex of both brain hemispheres. Regardless of the condition performed first, higher $\Delta P$ across frequency bands showed some characteristics that remained approximately stable for both task conditions. During HW, higher $\Delta P$ was observed at central brain areas, including the Primary Somato-Sensory Cortex (PSSC–white region in Fig. 4B), the premotor area, the medial parietal region related to the DMN (*Rojas et al., 2018*) (orange regions in Fig. 4B), and, mainly when starting by HW, areas of the frontoparietal control attention network (FPN) (*Japee et al., 2015*). The DMN is usually active during worries about the self. The higher $\Delta P$ at these regions could indicate a DMN and FPN co-activation. Worries due to an unconscious life-threatening situation, induced by the high-altitude, should promote an effortful focused attention on the "motor engram (memory trace)" for the whole-body motion (sensorimotor areas) (*Ueta et al., 2022*), to avoid falling during HW. LW showed higher $\Delta P$ on prefrontal sensors, for almost all frequency bands and situations (except when LW was first for the $\theta$ and $\alpha$ bands, Figs. 4A and 4B). Despite vmPFC being often associated to the DMN (*Corbetta & Shulman, 2002*; *Hänsel & von Känel, 2008*; *Rojas et al., 2018*), and commonly related to stress-relieving by counterbalancing amygdala activation (*Hänsel & von Känel, 2008*), the link between vmPFC and positive affective processing of safety signals has also already been shown (*Harrison et al., 2017*). Contrasting with the pioneer study considering EEG dynamics of flow during a tightrope performance (*Leroy & Cheron, 2020*), our findings do not corroborate the transient hypofrontality hypothesis (*Dietrich, 2003*) related with flow

experiences. Other noteworthy finding related to greater $\Delta P$ during LW was observed on electrodes positioned at the middle-frontal gyrus, at both hemispheres (Fig. 4B). This gyrus has been mainly linked to the stimulus-driven ventral attention network (VAN) (*Corbetta & Shulman, 2002*; *Harris, Vine & Wilson, 2017b*). Furthermore, higher $\Delta P$ during LW was also found for the occipital area, which is related to the brain's visual processing, as well as the starting area for visual streams pathways (*Goodale & Milner, 1992*; *Norman, 2002*; *Milner & Goodale, 2008*). Studies about cortical visual pathways suggest two streams (*Goodale & Milner, 1992*; *Norman, 2002*; *Milner & Goodale, 2008*): the ventral visual stream, directed to the temporal lobe (grey regions in Fig. 4B), and the dorsal visual stream, leading to the parietal lobe (purple region in Fig. 4B). The first is considered to be involved with objects' visual identification and recognition, having strong connections with the limbic system which controls emotions. LW showed some tendency for higher $\Delta P$ at the left on this stream. This tendency could indicate that flow experiences do not have lateralized right brain dominance, as reported to occur in effortful attention situations (*Coull, 1998*; *Lim et al., 2010*; *Nyberg & Cabeza, 2000*; *Ogg et al., 2008*; *Paus et al., 1997*), running discrete and distinct neural dynamics from task engagement situations, involving both hemispheres with a slight left dominance. The other visual stream, leading to the parietal lobe, is considered involved in processing spatial location with typically 'low self-consciousness'. The dorsal visual stream reaches the somatosensory association cortex, which interprets tactile sensory data being involved with the perception of limbs location in space, and the intraparietal sulcus, both areas considered to participate in the parietal brain region of DAN (*Japee et al., 2015*; *Lim et al., 2010*; *Ogg et al., 2008*) (parietal soft blue region in Fig. 4B). But, this stream also reaches the VAN at the inferior parietal lobe, more specifically at the supramarginal gyrus (CP5, CP6 electrodes), related to sensory information interpretation as well (*Carlson, 2012*) (posterior yellow cortex region in Fig. 4B). The VAN, also known as stimulus-driven network, can be responsible for a more opened attention to the surrounding details during a task, contributing to consider the moment-to-moment changes during action. Our current findings support an involvement of both dorsal and ventral attention networks relating to highest flow experiences.

**(H3) Flow experiences can arise as a cooperation of goal-directed and stimulus-driven brain attentional networks, with some $\Delta P$ prevalence for stimulus-driven mainly at frontal lobe.** VAN is mainly related to the middle frontal gyrus (MFG), inferior frontal sulcus (IFS) towards the inferior frontal gyrus (IFG), and to the temporo-parietal junction (TPJ) (*Japee et al., 2015*). DAN comprehends mainly the frontal eye fields (FEF), superior parietal lobule (SPL), and intraparietal sulcus (IPS) (*Japee et al., 2015*). Attempting to explain flow experiences in relation to brain networks activity, some studies consider a theory of the Multiple Demand (MD) network (*Harris, Vine & Wilson, 2017b*). According to this theory, the MD network overlaps VAN (*e.g.*, inferior frontal sulcus) and DAN (*e.g.*, intra parietal sulcus) (*Duncan, 2013*). As well as in STF (*Huskey, Wilcox & Weber, 2018*), the studies considering the MD network, further dividing attention brain networks into "sub-networks", suggest more "activation" of brain areas associated to the goal-directed DAN, with the 'optimal attentional control during flow' being dependent upon the absence of disruption and monitoring processes coming from the stimulus-driven VAN

(*Harris, Vine & Wilson, 2017b*). To verify this idea, we considered our results of high $\Delta P$ within the attention networks VAN and DAN, also divided into "sub-networks" at frontal or parietal-temporal regions, as 'collaborating regionally' (CR) or 'prevailing regionally' (PR) (Fig. 4B*)*. We further analyzed these variations concerning the relationship of the MD network attention hypothesis and our higher flow scores, *i.e.*, in LW. Nevertheless, note that our $\Delta P$ analysis approach could not determine if the activity of these two networks occurred in correlated synchronization or if it occurred in an oscillatory anti-correlated way (or, yet, in a desynchronization manner), but, since slackline can be pondered as a moment-to-moment task requiring constant attention resources to body balance, we presumed it pertinent and worthwhile to discuss the above assumptions. Thus, following this proposed approach, our results (mainly at frontal lobe) suggest that, besides VAN and DAN are collaborating to flow experiences, relevantly, rather than more attached to DAN, flow experiences would tend to have more significant prevailing of VAN (Fig. 4B) top and bottom rows. Although we did not perform eye tracking measurements, characteristically the slackliners tend to maintain a 'quiet eye' strategy directed at a horizontal point when walking the line. Some studies have associated "quiet eye" strategies with flow experiences (*Harris, Vine & Wilson, 2017a*). The frontal eye field (FEF) region is important in visual attention control and eyes' movement for seeking relevant targets or stimuli in the focus searching process (*Schall, 2004*). Perhaps, quiet eye strategies may possess a key role in DAN deactivation at the FEF area, and this decrease in FEF activation could allow the VAN system, mainly MFG/IFG region, to operate considering more incoming perceptions' details with no worries to find a focus of attention. Attention would then be calmly expanded to a higher awareness of the subject's surroundings, of the environment changing during the action, promoting the self-perception-performer-environment coupling (*Montull et al., 2020*).

VAN prevailing could solve the apparent paradoxical flow experiences descriptions which are difficult to conciliate when considering DAN predominance (*Harris et al., 2016*). Several flow reports emphasize 'enhanced feelings and perceptions' accompanying 'reduced attention effort' (effortless), contrasting at the same time with 'lack of distraction' and 'superior focus' on the task being performed. Recurring reports point to enhanced feelings and perception considerations during flow, even when apparently non-relevant to the task. A big-waves surfer describes improved perceptions of his surroundings, such as the wind, the details and variations in the face of the wave, and the salt in his mouth (*Partington, Partington & Olivier, 2009*). A vertical skateboarding rider mentions the feeling of being in total control, of every little movement, and very aware of everything, such as the rings in her hands (*Hunter & Csikszentmihalyi, 2000*). A free solo climber (climbing without safety ropes) discourses about how everything becomes elevated, with the awareness of the rock's texture, the surrounding beauty and birds whistling (*Sparks, 2016*). It seems more suitable to refer to enhanced consideration of perceptions, than to enhanced perceptions themselves, as these apparently non-task relevant perceptions were present the whole time-'salt in his mouth', 'rings in her hands', 'birds whistling'-, with

subjects remaining mostly unaware of them for the most part from normal sensory feedback during tasks. In this context, when in flow experiences, an increased amount of perceptions seems to ascend to consciousness, perhaps, turning the brain stimulus-driven network (VAN) very active, tending to improve the supply information for the brain sensory association area (DAN). With the leading role of the stimulus-driven VAN, the 'superior attentional focus' during flow, rather than commonly understood as an acute focused attention (*Csikszentmihalyi, 1990*; *Gold & Ciorciari, 2021*), can be understood as an opened attention to more perceptions coming from the surroundings. In this way, the subject's 'distractions disappear', because this superior, or, opened focus, is now handling an augmented quantity of inputs and stimuli constantly changing.

During flow, a person is often more open, alert, and flexible within the structure of the activity being performed (*Harris, Vine & Wilson, 2017b*). A heightened perception of the changings here and now tends to push to the present moment in a dynamical adaptive action and can be one of the catalyst that creates the opportunity to experience a flow state (*Csikszentmihalyi, 1978*). Action moments during flow are reported requiring to process several pieces of information in the environment simultaneously (*Wiersma, 2014*). Base jumpers describe flow as enabling to process more detailed sensory information. "Focus is opened in new or transcendent ways." "[…] you've got a kind of sensory overload […]" "I am fully aware of everything else that is going around me. It's the opposite of tunnel vision. […]" "You're at this level of alertness that you're not in a normal life." "[…] is a feeling of existential readiness." (*Brymer & Schweitzer, 2017*, p. 70).

Therefore, with the stimulus-driven network modulating the goal-directed decisions (*Eckert et al., 2009*), the 'optimal attention control during flow' would arise through the constant awakening to the dynamic adaptation opportunities. If so, flow would be denoted by a smooth continuous harmonically tuning and accurate optimal performance, overcoming habitual actions and embracing a more holistic experience.

Considering EEG graph analyses, the degree represents the amount of synchronization between a node (region) and other nodes (brain regions). The clustering coefficient reflects a similar notion, but more locally, in the sense that it evaluates the connections among a node's neighbors. These functional connectivity variations were presented on the significant electrode sites, through the size of the circles in Fig. 5. Due to the great relative metrics in HW, for normalized results, the lesser significant electrodes in LW appeared as small representative blue circles in proportion to the HW ones (Figs. 5A and 5B). Overall, for degree and CC, LW presented lower functional brain connectivity metrics values, and covered less regions with significant electrode sites compared to HW. In contrast to the STF, which conceptualizes flow as a cognitive network synchronization process optimizing brain functioning to handle a supposed 'limited capacity of attention', the desynchronization among ensembles of neural populations in our functional brain connectivity investigation can be interpreted as indicating that a de-coupling of brain regions better represents flow brain signatures, while overall synchronization of brain networks indicates higher negative emotion aspects with an inhibition of flow, rather than

its presence in HW. This could reflect a relaxation of constraints in global brain activity that corresponds with an expansion of the brain's possible repertoire (thereby increasing dynamic diversity) (*Kotler et al., 2022*). In our case, functional connectivity was estimated through coherence analysis. Other types of potential flow-inducing experiences, such as meditation, have also been studied using the coherence metric. A study with five different meditation traditions found lower significant coherence during meditation through all groups and all studied frequency bands (*Lehmann et al., 2012*). The study presumed that neither increase nor decrease of EEG coherence is a virtue in itself, given that general high coherence is observed during epileptic seizures while decreased coherence was found during schizophrenic symptomatology. Interestingly, availability of dopamine D2-receptors in the striatum were positively associated with schizophrenia (*Madras, 2013*), as well as with flow proneness (*de Manzano et al., 2013*). Anyway, since coherence assesses the cooperation between system units, lower coherence implies higher functional independence of the system units (*Lehmann et al., 2012*). This behavior, with functional brain regions independence, may reinforce the idea of an 'expanded model of attention' linked to flow, when compared to the normal everyday self-imposed limited attention. The EEG recordings disclosing the brain lighting up like a "fireworks display" can suggest more perception considerations coming from VAN, timely stimulating different brain areas by the moment-to-moment changes taking place on the overabundance of information around, contributing to the time perception distortion during flow (*Kotler et al., 2022*). A base jumper reports that "it is a complete sensation. It's a sensation that's taking in all your senses and then some that you didn't know you had" (*Brymer & Schweitzer, 2017*, p.68). She described it experiencing time "slowing down" with an increased ability to notice details in the environment, even while traveling at 125 miles (200 km) per hour:

> [...] you experience something interesting in that your awareness of 1 s expands enormously. What we would normally perceive in 1 s is very little compared to what you perceive in 1 s on a BASE jump. Your mind, so you can deal with everything that you have to, slows things down so when you're doing it, it feels like it's in slow motion. [...] When you watch it back on footage you look and go WOW, that's over in a blip (clicks fingers), but when you're doing it you know you can see the tiny little creases in the rock and different colors in the sky and you're totally aware of where your body is in space and how its moving and ... it's very surreal. (*Brymer & Schweitzer, 2017*, p. 69).

Theoretically, this brain pattern could cause the 'sense of readiness' to adapt and act by the sudden changes happening around, creating the gap for the 'action awareness merging feeling', the most characteristic feeling during flow. The merging feeling then, must generate the sensation of belonging to something greater, feature found in both meditative practices (*Lehmann et al., 2012*) and flow experiences (*Gold & Ciorciari, 2021*, *2020*; *Sparks, 2016*), reported as bliss, transcending, expanded consciousness, oceanic feelings, all-oneness, being changing together with the whole surroundings in a formlessness transient universe (*Huxley, 1956*).

This systemic fusion, with the self, expanded and integrated in the environment, becoming part of the same system, seems to fit well and characterizes a good consideration for the word/term labeled "'Flow'" by the theory pioneer (*Csikszentmihalyi, 1975*). One of

flow's most interesting phenomenological attributes is the sensation of flow itself, often described as effortless effort, where the experience is that every action and every decision being performed leads seamlessly, perfectly, fluidly to the next (*Csikszentmihalyi, 1990*; *Kotler et al., 2022*).

## CONCLUSION

Some proposed neurocognitive theories of flow experiences conjecture that there is a limited-capacity of available attentional resources and suggest that flow should result from the synchronization of the focused attentional control neural network, involving mainly the DAN. To support the role of increased modulation from DAN, these theories posit an inhibition of the salience-driven attention network that cover the VAN. Considering brain behavior linked to higher flow psychological subjective results, we found VAN prevailing regionally at the frontal lobe as well as overall cortex de-synchronization. Our interpretation of these results suggests flow experiences as an opened attention to more changing details in the surroundings, *i.e.*, configured as a 'task-constantly-opened-to-subtle-information experience', rather than a 'task-focused experience'.

This functional state of mind would enable a deep understanding of the self as part of a larger whole, promoting the 'action awareness merging feeling', raising a shared identity with all universe conjoint motion, in the instance which the 'individual member' shares dynamic and harmonic action with everything around, improving harmonic performance as consequence. These highly synergistic dynamic experiences, fashioning jointly with the environment, match well with extreme sports in which athletes often describe the action as a partnership dance, expressing it as feeling part of nature, with enhanced senses and altered perceptions of time and space, including new ways of perceiving the body-space continuum, as if every cell was twitching and alive (*Brymer & Schweitzer, 2017*). These glimpse regards can raise an expanded and enlightening consideration for the concept that everything is relative. Everything is relative only because everything is the same thing: 'movement'. By 'movement', everything influences everything else, while being influenced by all everything. The 'action awareness merging flow experience insight', or, the sensation of belonging, sharing oneness feeling with the 'whole movement', which actually permeates the 'entire universe', provide peace, calm and stillness during motion.

As practical implications, the findings of the present study provide a more comprehensive explanation of human perception during flow in extreme sports performance and suggest important insights regarding the possibility of temporal processing and spatial attention. Despite the difficulties in using sensitive equipment outside of the laboratory as well as the logistical problems, our approach attempted to emulate extreme sports' real-world conditions. As equipment development becomes more robust to noise or waterproof, it will encourage more attempts to use extreme sports such as high-altitude slacklining, free solo climbing, wingsuits proximity flying, or giant waves riding, to study physiological and neuro-cognitive correlates of more complex and intense real experiences of flow. Improvements in equipment, allowing enhanced patterns seeking while studying flow experiences in extreme sports, could have interesting practical

applications not only for extreme sports. One idea could be to develop trainings focused on promoting flow while tracking brain oscillations in traditional sports or in elite military troop missions. In these kinds of scenarios, turning on the 'sense of readiness' by adapting action according to the sudden changes happening around, may promote the 'action awareness fusion'. This feeling comprises influence waves emerging from the self toward surrounding events, while perceiving actions happening in slow motion, which might, *per se*, abruptly increase performance.

## Limitations

Approaches such as ours to study brain networks inherently oversimplify the true brain network topology, where multiple regions should be functionally connected (*Huskey, Wilcox & Weber, 2018*), or where an 'over request' of one or more brain network regions influence the others; hence results should be considered with some caution. For example, our study did not involve sub-cortical interactions and it also over-simplified analyses considering the cortical areas underneath positioned electrodes. Nevertheless, we found a general brain-cortex desynchronization relative to subjective flow experience. The choice of functional connectivity measure, namely, coherence, was due to the wide use of this measure in the EEG literature; this measure has thus been extensively investigated and validated in several studies (*Srinivasan et al., 2007*). So, we took the opportunity to make relations with invaluable works that used the coherence approach, (*e.g.*, *Lehmann et al., 2012*), what led us to rise interesting considerations comparing our results with other important studies (*de Manzano et al., 2013*; *Madras, 2013*). Although it may be argued that coherence suffers from volume conduction artifacts, the use of CAR (common average referencing) (*Ludwig et al., 2009*) in the preprocessing stage should decrease this possibility (*Brunner et al., 2016*; *Yuksel & Olmez, 2016*). The use of eye-tracking equipment could have further clarified our results, especially $\Delta P$ results at FEF area in the frontal cortex, but unfortunately we did not have access to this type of equipment at the time of data collection. The results dependence on the conditions' order was certainly influenced by the lack of training-adaptation periods before the measurements (data collect): starting in LW might have served as a psychological preparation for the task on HW, and *vice-versa*. Nevertheless, the lack of such training-adaptation sessions was necessary in order to include as many participants as possible; in our case, eight (four on Saturday and four on Sunday). In this respect, several factors influenced on or reduced our sample size. For example, travelling to the measurement site, preparing participants with the equipment and the tasks performance itself demanded extensive time. Moreover, obtaining the appropriate authorization to use the equipment outside the laboratory environment was complicated, so that the authorization was granted only for gathering data over 1 weekend. On the other hand, this approach brought our investigation closer to real-world scenarios. In addition, all our analyses relied on powerful statistical modelling and/or seeking common trends for all subjects. Therefore, whereas the small number of subjects might preclude generalization of our conclusions, allied to the fact that this is an exploratory study which needs future works to better affirm the strength of our raised hypotheses, we are providing, to the best of our knowledge, besides the initial work evaluating one

tightrope highline walking (*Leroy & Cheron, 2020*), an early support for seeking the corresponding psycho-physio-neurological correlates of brain behavior close to the real life in extreme sports, as well as an innovative description for this type of data, something that can certainly be replicated, improved and augmented in future studies on this research area.

## ACKNOWLEDGEMENTS

We would like to thank the Centre de Santé ROSSETI for the partnership in sharing the used EEG equipment, Marion Fournier for her help with data collection, Stephen Ramanoel for his later valuable comments and advices that helped to improve this study and Raphael Zory for his help with requested documents after Dr. Radel's decease. Requests on this article can be sent to Gabriela Castellano, rua Sérgio Buarque de Holanda, 777, Campinas 13083-859, Brazil, gabriela@ifi.unicamp.br.

### Funding

Marcelo Felipe de Sampaio Barros was supported by a scholarship from the CAPES Foundation of the Ministry of Education of Brazil, Brasília–DF 70040-020, Brazil (number 99999.010663/2014-02). Rémi Radel was supported by a grant of the Agence Nationale de la Recherche (ANR–JCJC, 2013-069). This research was also supported by CAPES Foundation of the Ministry of Education of Brazil-Finance Code 001, São Paulo Research Foundation-Brazil (FAPESP)-Grant #2013/07559-3, and Brazilian National Council for Scientific and Technological Development (CNPq)-Grant #304008/2021-4. The funders had no role in study design, data collection and analysis, decision to publish, or preparation of the manuscript.

### Grant Disclosures

The following grant information was disclosed by the authors:
Ministry of Education of Brazil, Brasília–DF 70040-020, Brazil: 99999.010663/2014-02.
Agence Nationale de la Recherche: ANR–JCJC, 2013-069.
 Ministry of Education of Brazil-Finance Code 001, São Paulo Research Foundation-Brazil (FAPESP): #2013/07559-3.
Brazilian National Council for Scientific and Technological Development (CNPq): #304008/2021-4.

### Competing Interests

The authors declare that they have no competing interests.

### Author Contributions

- Marcelo Felipe de Sampaio Barros conceived and designed the experiments, performed the experiments, analyzed the data, prepared figures and/or tables, authored or reviewed drafts of the article, and approved the final draft.

- Carlos Alberto Stefano Filho analyzed the data, prepared figures and/or tables, authored or reviewed drafts of the article, and approved the final draft.
- Lucas Toffoli de Menezes analyzed the data, prepared figures and/or tables, and approved the final draft.
- Fernando Manuel Araújo-Moreira conceived and designed the experiments, authored or reviewed drafts of the article, and approved the final draft.
- Luis Carlos Trevelin conceived and designed the experiments, authored or reviewed drafts of the article, and approved the final draft.
- Rafael Pimentel Maia analyzed the data, prepared figures and/or tables, authored or reviewed drafts of the article, and approved the final draft.
- Rémi Radel conceived and designed the experiments, performed the experiments, analyzed the data, authored or reviewed drafts of the article, and approved the final draft.
- Gabriela Castellano analyzed the data, prepared figures and/or tables, authored or reviewed drafts of the article, and approved the final draft.

**Human Ethics**

The following information was supplied relating to ethical approvals (*i.e.*, approving body and any reference numbers):

CER–Comité dÉthique de la Recherche d'Université Côte d'Azur

**Field Study Permissions**

The following information was supplied relating to field study approvals (*i.e.*, approving body and any reference numbers):

The study's protocol was approved by the scientific council of the University (CER–Comité dÉthique de la Recherche d'Université Côte d'Azur), under the supervision of professor Rémi Radel, Lecturer at the University of Nice Sophia Antipolis Côte D'azur, and under responsibility of professor Raphael Zory, director of Laboratoire Motricité Humaine Expertise Sport Santé-LAMHESS

**Data Availability**

The data is available at Zenodo: de Sampaio Barros, M. F., Stefano Filho, C. A., Toffoli de Menezes, L., Araújo-Moreira, F. M., Trevelin, L. C., Pimentel Maia, R., Radel, R., & Castellano, G. (2023). Psycho-physio-neurological correlates of qualitative attention, emotion and flow experiences in a close-to-real-life extreme sports situation: low- and high-altitude slackline walking [Data set]. Zenodo. https://doi.org/10.5281/zenodo.8172597.

**Supplemental Information**

Supplemental information for this article can be found online at http://dx.doi.org/10.7717/peerj.17743#supplemental-information.

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
