# Peer review of "Psycho-physio-neurological correlates of qualitative attention, emotion and flow experiences in a close-to-real-life extreme sports situation: low- and high-altitude slackline walking"

_PeerJ, doi:10.7717/peerj.17743_

## Round 0.1 · original submission · Minor Revisions

The aim of the present study was to investigate changes in the psychological, peripheral-physiological in response to a high-risk sport situation.
While this study contains interesting data, reviewers have highlighted the need for some modifications before considering it for publication. Please review the comments provided by two reviewers and respond to each point accordingly.

Reviewer 1 ·

Basic reporting

no comment

Experimental design

no comment

Validity of the findings

no comment

Additional comments

The aim of the present study was to investigate changes in the psychological (qualitative attention, emotion and subjective flow experiences), peripheral-physiological (heart rate and breathing rate), and neuro-physiological (EEG analysis) in response to a high-risk sport situation, such as slacklining at low and high altitude. The study is well-written, easy to read, and provides new insights into the psychophysiological correlates of practicing an extreme sport.
The introduction offers a thorough examination of the research field related to extreme sports, with a specific emphasis on the psychophysiological reactions connected to engaging in high-risk sports. The authors adeptly emphasize the changing comprehension of extreme sports, moving beyond the focus on individuals seeking intense sensations and instead highlighting the intricate nature of the experiences involved in these activities. Moreover, the introduction establishes a distinct justification for the study by pinpointing deficiencies in existing knowledge, specifically concerning the neurophysiological foundations of extreme sport experiences.
The findings suggest that participants in low-walk conditions had higher scores on the flow scale, while those in high-walk conditions had higher scores in negative affect aspects, suggesting possible limitation of flow during high-walk conditions. Moreover, the order in which tasks were performed had a significant impact on physiological and neural variables, indicating a sort of confidence based on previous experience. Neurophysiologically, flow experiences involve a broad and receptive focus on subtle cues in the environment, rather than being solely focused on the task per se. Figures and Results are presented with high-quality.
Discussion is clear and provides and in-depth analysis of the findings in light of the literature. Overall, I would like to congratulate the Authors for their work that deserves publication.


I have only a couple of minor suggestions that I hope can contribute to complete the overall quality of the manuscript.
In the Discussion section, high heart rate is considered as a proxy of negative emotions. I would suggest to briefly discuss possible future perspectives on the assessment of emotions based on Machine Learning approach. Here a suggested reference to refer:
Filippini, Chiara, et al. "Automated affective computing based on bio-signals analysis and deep learning approach." Sensors 22.5 (2022): 1789.
Moreover, I do believe that including possible practical applications derived from these findings might be helpful for the readers and also for practitioners working with individuals practicing extreme sports (such as slacklining, free climbing, mountaineering…). What are the practical implications and applications that can be derived from these findings?

·

Basic reporting

It is a very interesting topic and difficult to do research on since the brain is highly fluctured in an unconstrained environment. I have some detailed feedback and some overall ideas to make the manuscript more clear to the readers:
line 90: please find a more up to date reference as you want to proof that the extreme sports are nowadays popular (and not in 2001 or 2012).
Line 131: the reference B: should be Brian Bruya as the author. (Bruya, 2010)
135-137: Surgery and mountain climbing are highly critical tasks, which are more often reported to result in intense, ecstatic flow experiences, whereas yet absorbing but less critical tasks such as reading and video games, have less intense flow feelings (Gold & Ciorciari, 2020): I think these two types of flow cannot be seen on one 'scale' going form less intense flow to intense flow. Is this the same 'flow theory' in the Gold article as Csikszentmihalyi? In my opinion, a person can experience intense flow also during a video game (for example in a gamer expert doing a new game), It depends on the person and his activity. I did not read the article of Gold, maybe they say it like that, but in my experience of 'flow', this should be more nuanced.
line 161; this is also an interesting reference: Yu D, Wang S, Song F, et al. Research on user experience of the video game difficulty based on flow theory and fNIRS. Behaviour & Information Technology. 2022:1-17.
from line 181 to the end of the introduction: this is valuable information, however, some lines feel to be in the method part or even in the discussion part of the article.

Experimental design

The method section is detailed written and transparant.
402: you used the 30s rest period as a baseline, however, in the introduction you mentioned that the time before the activity is already stressful, is the 30sec rest before the start of the activity a non-biased baseline point then?
The experiment is rigid and trustable, however, eight participants are not enough to conclude in general, but I understand the small group due to the specific conditions (and is also well written in the limitation part).

Validity of the findings

The result section is very precised written, well done. The discussion linked the results with the existing theories. It would be a strenght to more highlight the novality of the study and the main findings, it was difficult to following the results and discussion since there was an overload of data.

Additional comments

The first sentence in the abstract: "It has been indicated that extreme sport activities, despite providing fear, stress and anxiety, also result in a highly rewarding experience." reflects the extreme sport as negative since you sum up the fear stress and anxiety, I would suggest to delete this or reframe this, is it relevant to state the fear, stress and anxiety factor as the first things to know when you introduce extreme sport activities? I would say : the adrenaline, the kick... (however, in the introduction line 75-79; it is more clear why you relate these two things, so please rephrase that it is more clear)
341; reference has a letter that does not fit
Please check the references again in the text and in the list.

---

## Round 0.2 · accepted · Accept

The editor has confirmed that the authors have appropriately revised the manuscript in accordance with the reviewers' comments.